

# A New Global Gridded Sea Surface Temperature Data Product Based on Multisource Data

Mengmeng Cao[1†], Kebiao Mao[1, 2, †], Yibo Yan[1], Jiancheng Shi[3], Han Wang[1], Tongren Xu[4], Shu Fang[5], Zijin Yuan[1]

[1]Institute of Agricultural Resources and Regional Planning, Chinese Academy of Agricultural Sciences, Beijing, 100081, China
[2]School of Physics and Electronic-Engineering, Ningxia University, Yinchuan, 750021, China
[3] National Space Science Center, Chinese Academy of Sciences, Beijing, 100190, China.
[4] State Key Laboratory of Remote Sensing Science, Jointly Sponsored by the Aerospace Information Research Institute of Chinese Academy of Sciences and Beijing Normal University, Beijing, 100101, China.
[5] School of Earth Sciences and Resources, China University of Geosciences,Beijing, 100083, China
*Correspondence to:* Kebiao Mao (maokebiao@caas.cn)
†These authors contributed equally to this work and should be considered co-first authors.

**Abstract:** Sea surface temperature (SST) is an important geophysical parameter that is essential for studying global climate change. Although sea surface temperature can currently be obtained through a variety of sensors (MODIS, AVHRR, AMSR-E, AMSR2, Windsat, in situ sensors), the temperature values obtained by different sensors come from different ocean depths and different observation times, so different temperature products lack consistency. In addition, different thermal infrared temperature products have many invalid values due to the influence of clouds, and passive microwave temperature products have very low resolutions. These factors greatly limit the applications of ocean temperature products in practice. To overcome these shortcomings, this paper first took MODIS SST products as a reference benchmark and constructed a temperature depth and observation time correction model to correct the influences of the different sampling depths and observation times obtained by different sensors. Then, we built a reconstructed spatial model to overcome the effects of clouds, rainfall and land interference that makes full use of the complementarities and advantages of SST data from different sensors. We applied these two models to generate a unique global 0.041° gridded monthly SST product covering the years 2002–2019. In this dataset, approximately 25% of the invalid pixels in the original MODIS monthly images were effectively removed, and the accuracies of these reconstructed pixels were improved by more than 0.65°C compared to the accuracies of the original pixels. The accuracy assessments indicate that the reconstructed dataset exhibits significant improvements and can be used for mesoscale ocean phenomenon analyses. The product will be of great use in research related to global change, disaster prevention and mitigation and is available at http://doi.org/10.5281/zenodo.4419804 (Cao et al., 2021).

## 1 Introduction

The temperature at the interface between the atmosphere and ocean, known as the sea surface temperature (SST), is an





important indicator of Earth's ecosystem (Hosoda and Sakaida, 2016). SSTs are widely used in atmospheric and oceanographic studies, such as in atmospheric simulations, climate change monitoring, and in studies of marine dynamic environments (Kawai and Wada, 2007; Martin et al., 2007; Peres et al., 2017; Reynolds and Smith, 1995). In addition, the oceans cover 70% of Earth's surface. A small variation in the ocean temperature exerts strong impacts on regional and even global climate change, energy exchange and the environment due to the unique physical characteristics of the oceans, including their high heat capacity (Varela et al., 2018). The rise of ocean temperatures will release huge amounts of heat, affect atmospheric movement, and produce many chain reactions, causing reductions in the $CO_2$ content of seawater, the occurrence of extreme weather, the melting of sea ice in the polar region, and the rise of sea level, all of which will impact the survival of marine life, marine production and human life (Sakalli and Basusta, 2018). Thus, it is essential to accurately monitor changes in SST.

It is difficult for traditional SST measurements based on buoys, platforms and voluntary ships to obtain large-scale and synchronous SST data due to the large gaps present in the data over both space and time. Compared to the traditional in situ SST monitoring approach, remote sensing technology has advantages in terms of large-scale and dynamic monitoring and has been used to acquire global ocean SST observation data (Li and He, 2014). Satellite SST data include infrared and microwave radiometer SST data. Retrievals from satellite thermal infrared sensors can provide global SSTs at high temporal frequencies and spatial resolutions of typically 1-4 kilometers with low uncertainty (Alerskans et al., 2020). For example, series sensors such as the Moderate Resolution Imaging Spectroradiometer (MODIS) and Advanced Very High Resolution Radiometer (AVHRR) can measure global SSTs with high resolutions and high accuracies. These observations are unfortunately greatly influenced by the atmospheric environment. In cases of aerosol contamination and cloud cover, it is impossible to obtain effective observations, resulting in spatial discontinuities and low quality in the collected data (Guan and Kawamura, 2003; Hosoda et al., 2015; Liu et al., 2017b). In contrast to infrared measurements, microwave sensors are less affected by clouds and aerosol concentrations (Alerskans et al., 2020). Therefore, microwave sensors can observe SST information at all times and in all weather conditions except rain, and they also have high temporal resolutions and can quickly cover the whole surface of Earth (Wentz et al., 2000). As a result, microwave sensors play important roles in monitoring the temporal and spatial changes in SSTs on global and continental scales and have also been developed into mature remote sensing products, such as the TRMM Microwave Imager (TMI), the WindSat on-board Coriolis, and the Advanced Microwave Scanning Radiometer for Earth Observation System (AMSR-E), which have been widely used to retrieve SSTs (Gentemann, 2014; Ng et al., 2009; Purdy et al., 2006). However, the spatial resolutions of passive microwave sensors are very coarse and are greatly affected by land and sea surface wind and waves, which makes it impossible to obtain detailed information about SSTs (Gentemann et al., 2010; Liu et al., 2017a). In addition, due to the influence of imaging orbit gaps, microwave-based products produce spatial gaps. Therefore, the SST information obtained by a single satellite remote sensor is often incomplete and limited and cannot fully meet the user's demand for a dataset with a high resolution, high precision and full spatiotemporal coverage. Luckily, the simultaneous availability of multiple satellite sensors provides highly complementary information, enabling the production of high-quality unified SST datasets with improved global



coverage (Guan and Kawamura, 2004; Shi et al., 2015; Thiebaux et al., 2003).

Many SST fusion algorithms use multiple satellites and in situ data to take advantage of the strengths of each SST observation and solve the above issues; these algorithms include objective analysis (OA), optimal interpolation (OI), three-dimensional variational (3-D Var), and Kalman filtering (KF) (Chao et al., 2009a; Li et al., 2013; Smith and Reynolds, 2003). Bretherton et al. (1976) first applied OA in a study of ocean data. OI was developed on the basis of OA, and in OI, background information is introduced in the analysis process. Although there is no physical constraint, the OI has a perfect mathematical form, which statistically takes into account the influence of the relative position changes of different observation points on the error covariance. The OI algorithm is simple and easy to use and has become one of the main methods currently used for SST fusion. For example, Reynolds and Smith (1994) used the OI method to fuse in situ data from ships, buoys and satellites to produce OISST products that are widely used. The other SST analysis data product, RTG-SST from the National Centers for Environment Prediction (NCEP), are also obtained by the OI method. In addition, based on the Modular Ocean Model (MOM), the National Science Foundation and the National Oceanic and Atmospheric Administration established the Simple Ocean Data Assimilation system (SODA) by the OI method (Carton et al., 2018; Carton and Giese, 2008). Based on the Modular Ocean Model version 4p1 (MPM4), the Australian Bureau of Meteorology established marine forecasting systems covering Australia, nearby regions and the globe through the EnOI method (Oke et al., 2008). However, in practice, to reduce the computational burden, the OI algorithm is usually only applied using data near the analysis point, and there is often a certain degree of subjectivity. Methods such as VAR and KF have been proposed to overcome these problems, and these methods have been widely used. For example, Zhu et al. (2006) developed a new 3DVAR-based Ocean Variational Analysis System (OVALS), which can effectively improve estimations of temperature and salinity by assimilating various observed data. Li et al. (2008) applied a new 3D VAR data assimilation scheme to a retroactive real-time forecast experiment, and favorable results were obtained. In terms of operational applications, some institutions in Canada, the United Kingdom, the United States and China have used this method to establish ocean environmental forecast and analysis systems based on different oceanic general circulation models (Burnett et al., 2014; Chassignet et al., 2009; Han et al., 2011; Storkey et al., 2010). Huang et al. (2008) filled in the missing parts of satellite SST data with the kriging interpolation method based on the slowly changing characteristics of SSTs and then used KF to coordinate the variation error and interpolation error of the obtained SSTs. Finally, the interpolation and filtered SST data were fitted to realize SST filling. Wang et al. (2010) used the KF method to fuse the AVHRR SST and AMSR-E SST products to produce daily, spatially continuous SST data with a spatial resolution of approximately 2 km. However, a daily variation correction was not carried out before the fusion, and the model processing error was not taken into account, which brought great uncertainty to the fusion results.

The above research has greatly improved the accuracy and spatial coverage integrity of SST products, and a variety of SST fusion products have been generated that have excellent accuracy in deep water regions (Dash et al., 2011). However, there are also important deficiencies in some SST fusion products. On the one hand, the SST observations obtained by different sensors are highly complementary, but there are certain differences in SST products from different sensors because



different sensors can effectively respond to water column temperatures at different times and depths (Castro et al., 2004; Wick et al., 2004). Despite the use of various technologies to blend multiple SST products after rigorous quality control at each datapoint, the differences still need to be resolved. On the other hand, although these SST products have high accuracies for the global ocean, some products have problems with missing pixels and relatively low accuracies near coasts
and the edges of sea ice due to the characteristics of the remote sensing products themselves and the insufficiencies of fusion methods (Xie et al., 2008). Last, the assimilation and fusion products of multisource oceanic data can solve state estimations of oceanic large-scale ocean phenomena well, but some of these products cannot meet the needs of near-shore or small- and medium-scale phenomena. To address the above issues, we constructed a temperature depth and observation time correction model to eliminate the sampling depth and temporal differences among different data. Then, we proposed a reconstructed
spatial model that filters out missing pixels and low-quality pixels from the monthly MODIS SST dataset and reconstructs them based on daily in situ SST data and daily satellite SST retrieval data from two infrared (MODIS and AVHRR) and three passive microwave (AMSR-E, AMSR2, Windsat) radiometers to generate a high-quality unified global SST product with long-term spatiotemporal continuity. The dataset that takes advantage of complementarities and advantages of SST data from different sensors has a 0.041° grid of monthly observations covering the years 2002–2019 and was validated and cross-
compared with in situ observations and other SST products. The results indicate that the new reconstructed SST data is reliable and is suitable for regional or global SST studies.

## 2 Data and methods

### 2.1 Satellite data retrievals

Infrared and microwave radiometers on sun-synchronous satellites are the primary technical tools used to obtain global SST,
and collectively, these sensors provide highly complementary information with which a new SST product can be generated. The AVHRR and MODIS satellites, which cover the global ocean, were selected as sources of infrared radiometer data. To reduce the data gaps present in infrared data resulting from cloud and water vapor contamination, the inclusion of microwave radiometer data from polar-orbiting satellites are essential; in this study, ASMR-E, WindSat and AMSR2 are the main sources of microwave data.
The MODIS sensor is onboard the Terra and Aqua spacecraft: the sensor has an ascending local equatorial crossing time of 13:30 in the case of the Aqua spacecraft and a 10:30 descending equatorial crossing time for the Terra spacecraft. The daily and monthly L3m global SST products (Day and Night) of the MODIS sensor from Terra and Aqua are available starting from February 2000 and July 2002, respectively, with a 0.041° spatial resolution; these datasets were mainly used to reconstruct high-quality SST data and are available through the website https://oceandata.sci.gsfc.nasa.gov/. The standard
deviation obtained in a data comparison was better than 0.43°C, as determined by comparison of the SST data with coincident ferry observations (Barton and Pearce, 2006). Each pixel of these SST data is associated with a numerical quality level stored in SST_flags whose value ranges, in order of descending quality, from 0 to 4. Clear data of the best quality are





limited to the satellite zenith angles, < 55 degrees. Clear pixels at satellite angles > 55 degrees have good quality, with quality levels of 1. Pixels with a quality level > 1 may have very large differences between the retrieved SST and the

reference SST due to significant cloud contamination or various other problems (https://oceancolor.gsfc.nasa.gov/atbd/sst/). Therefore, these pixels are not used for scientific research.

The AVHRR sensor is onboard NOAA polar-orbiting satellites, has 6 bands ranging in wavelength from visible to infrared (one visible, two near-infrared, and three thermal infrared) and can cover the globe twice a day. The twice-daily (Day and Night) AVHRR 4-km SST data product is produced by the NOAA National Centers for Environmental Information and is

available through the website https://data.nodc.noaa.gov/pathfinder/Version5.3/L3C/. The standard deviation obtained in a data comparison is approximately 0.68°C, as determined by a comparison of the AVHRR products with coincident ferry observations (Barton and Pearce, 2006). The data also provided a quality index for each pixel based on the evaluation test results stored in the pathfinder_quality_level metric, which allows the identification of cloudy pixels and/or suspicious observations, with the quality level 0 representing the worst quality and the quality level 7 being the best (Pisano et al., 2016).

In our data processing method, we only considered values with quality flags 4 ~ 7.

The AMSR-E sensor is onboard the Aqua satellite and is a dual-polarization microwave scanning radiometer with 6 frequency channels in the range of 6–89 GHz. The AMSR-E instrument was in orbit for nearly 10 years but was discontinued in October 2011, owing to an antenna rotation problem. The AMSR2 sensor, onboard the Global Change Observation Mission-Water 1 (GCOM-W1) satellite, was launched in May 2012 to continue the Aqua/AMSR-E

observations and ensure the continuity of SST data (Zabolotskikh et al., 2015). AMSR2 has the same channels as did AMSR-E, with a 7.3-GHz channel added to help alleviate radio frequency interference. However, SST information collected from the AMSR2 sensor was not provided until mid-2012. To ensure that there is an uninterrupted consistent long-term microwave SST time series that can be used to reconstruct a high-quality SST product, a WindSat polarimetric radiometer was used to bridge the gap between the AMSR-E and AMSR2 products. The daily L3 SST products (ascending and

descending passes) of AMSR-E and AMSR2, available from June 2002 and July 2012, respectively, with 0.1°-grid spatial resolutions, were used to reconstruct high-quality SST data and are available through the website https://gportal.jaxa.jp/gpr/search/. The accuracies of AMSR-E and AMSR2 are approximately 0.75°C and 0.56°C, respectively, as determined by comparisons with buoy data (Sun et al., 2018). Daily WindSat SST datasets on a global 25-km grid (ascending and descending passes) were downloaded online (http://www.remss.com/missions/windsat), and their

accuracies are very close to that of AMSR-E, as determined by comparisons with buoy data (Banzon and Reynolds, 2013; Gentemann, 2011).

**2.2 In situ observations**

In situ observations of SST from 2002-2019 were used for the reconstruction of the new SST product and the validation of both the satellite-obtained SST data and the new product. The in situ observed SST data used in this study consist of SSTs

from the Version 2.1 NOAA in situ Quality Monitor (iQuam), which includes updated observations every 12 hrs with a 2-hr





latency. The SST data from iQuam include observations from drifters, ships, tropical (T-) and coastal (C-) moorings, agro floats, high resolution (HR) drifters, IMOS ships, and coral reef water (CRW) buoys, and the data can be obtained from ftp://ftp.star.nesdis.noaa.gov/pub/sod/sst/iquam/v2.10/. Quality control of the data, including basic screening, duplicate removal, plausibility, platform tracks, referencing, cross-platform and SST spike checks, was performed by the NOAA
Center for Satellite Application and Research (Xu and Ignatov, 2014). Only SSTs assigned the best quality flag (i.e., level 5) were used in this study. To ensure the independence of the data reconstruction and the accuracy verification process, all spatially coincident daily iQuam SSTs with temporal sampling less than or equal to 1 hr were used to reconstruct the MODIS SST data, while spatially coincident monthly SSTs calculated from daily SSTs were used to verify the accuracy of the reconstruction results (Minnett, 1991). The spatially coincident criterion restricts the maximum distance between in situ
measurements and the center of the satellite image grid cells to within 2.3 km, which is approximately half the spatial resolution of MODIS, so that the in situ observations always fall within the MODIS SST pixels (Pisano et al., 2016).

## 2.3 Ancillary data

ERA-Interim, a climate reanalysis product produced by the European Centre for Medium-Range Weather Forecasts (ECMWF), was discontinued on 31 August 2019 and has been superseded by the ERA5 reanalysis product produced
by the ECMWF. The ERA5 dataset is the latest climate reanalysis product, providing hourly data on atmospheric, land and oceanic climate parameters together with estimates of uncertainty. The 10-meter wind component U, 10-meter wind component V, 2-meter temperature, 2-meter dewpoint temperature, sea surface temperature, relative humidity, cloud cover and other data from the two datasets with 0.25° spatial resolutions were used to calculate the heat, momentum and freshwater fluxes between the ocean and the atmosphere as well as the incoming solar radiation. These
data can be obtained from https://apps.ecmwf.int/datasets/.

## 2.4 SST data development

Since MODIS SST data have a high accuracy and spatiotemporal resolution and can be used to capture mesoscale phenomena in the oceans, combining MODIS SSTs from Aqua and Terra is a good way to improve the spatial coverage of SST data. However, infrared SST data are retrieved using the infrared band, which cannot penetrate clouds, so SST data
cannot be provided in the presence of clouds. Furthermore, infrared SST retrievals are greatly influenced by atmospheric aerosols and water vapor. Some factors related to radiometers can also contaminate SST observations, such as the viewing geometry, spectral response, and noise level of each sensor (Kilpatrick et al., 2015). Due to these factors, MODIS SST data often have problems involving low-quality or missing pixels. Statistical analysis performed during the study period indicated that the unusable pixels present in the monthly SST records of Terra and Aqua during both daytime and nighttime generally
cover 23.46% and 28.06% of the global ocean, respectively. It is difficult to fill the data gaps in the MODIS SST retrievals caused by the above factors using infrared SST retrievals with data of the same quality as SST measurements collected under



clear sky. Therefore, we built a reconstructed spatial model that combines in situ station-based data and daily SST data from AVHRR, AMSR-E, AMSR-2 and WindSat to reconstruct a high-quality monthly MODIS SST dataset that takes into account the actual SSTs under clear-sky conditions instead of under clouds. More details are given in the following sections.

In addition, there are certain biases present in SST products due to the use of different sensors resulting from the measurement methods, sensor band settings, and environmental influencing factors in both space and time. Thus, after identifying and accounting for these differences, we proposed a temperature depth and observation time correction model to address the influence of time phase and sampling depth of different sensors. The overall methodology is illustrated in Figure 1. The processing effectively retains the pixels with high accuracy in the original MODIS daily and monthly data, and uses

ocean multisource data after calibration by using the temperature depth and observation time correction model and combines the spatio-temporal information to reconstruct the low-quality and missing daily pixels, and finally replaces low-quality and missing pixels with the composite average pixel value in the monthly data.

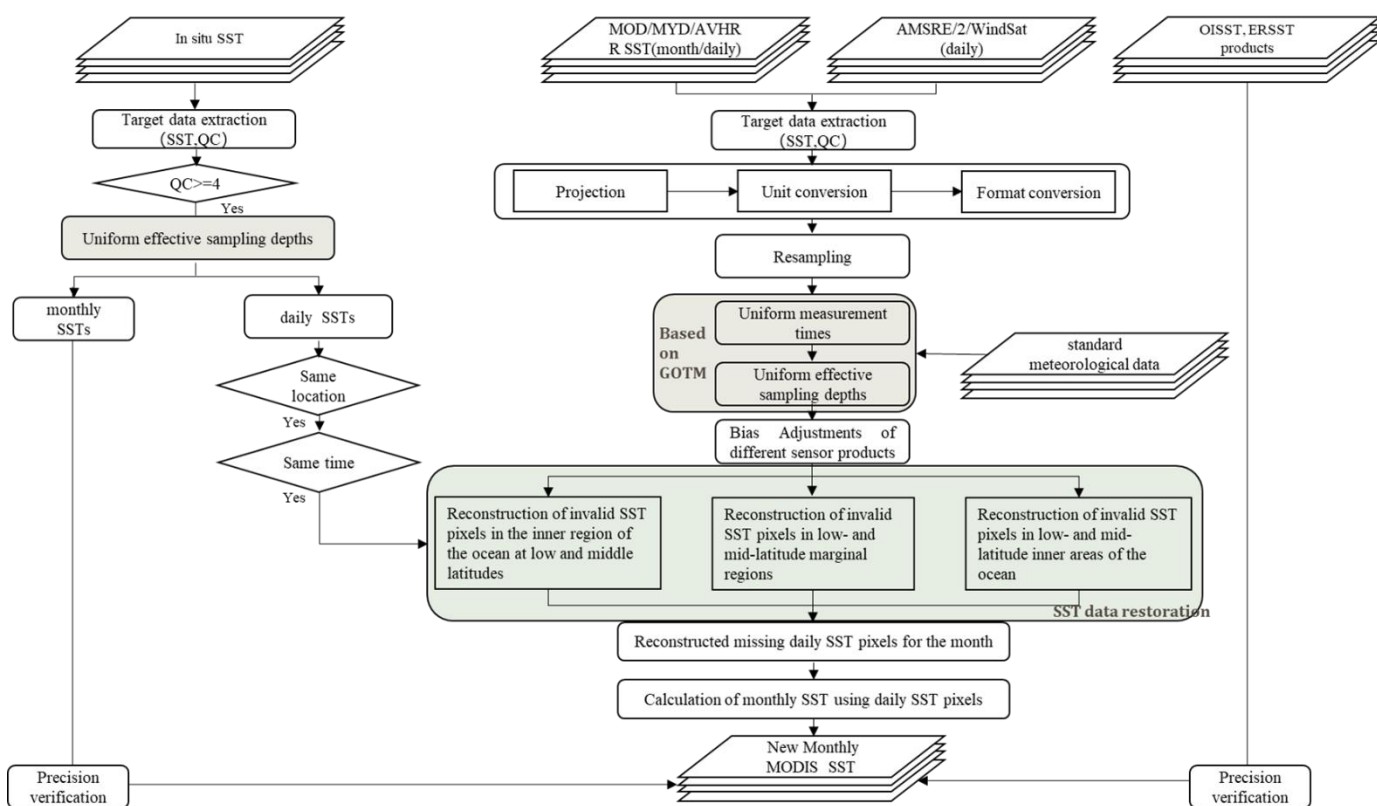

Figure 1. A summary flowchart for reconstructing MODIS monthly SST data

**2.4.1 Bias adjustment schemes**

**2.4.1.1 Bias adjustment scheme for multisource remote sensing data**

To combine oceanic multisource remote sensing data into the MODIS SST product, it is necessary to assume that the



measured values represent the same quantities or to use some method to eliminate the differences among products. The ocean temperature data obtained by different sensors are different from those obtained by MODIS, and there are complex

spatiotemporal differences among the sensors. Figures 2 and 3 represent the different distributions of the original MODIS and multisource daily SSTs in the daytime. Obviously, these multisource data cannot be directly used to reconstruct the valid pixels of MODIS SST data before the differences are corrected.

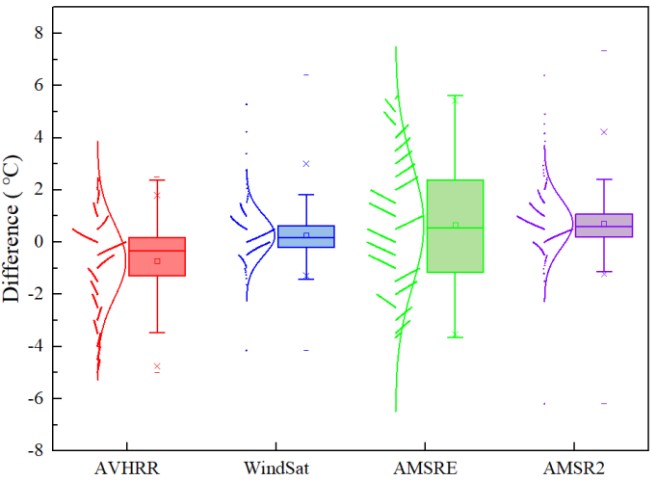

Figure 2. Box chart with scatters of the differences in the original MODIS and multisource daily SSTs (AVHRR, WindSat,

AMSRE, AMSR2). The boxes are determined by the 25th and 75th percentiles. The whiskers are determined by the 5th and 95th percentiles. The data are plotted as scatters on the left of each box. A curve corresponding to a normal distribution is also displayed on top of each scatter plot.

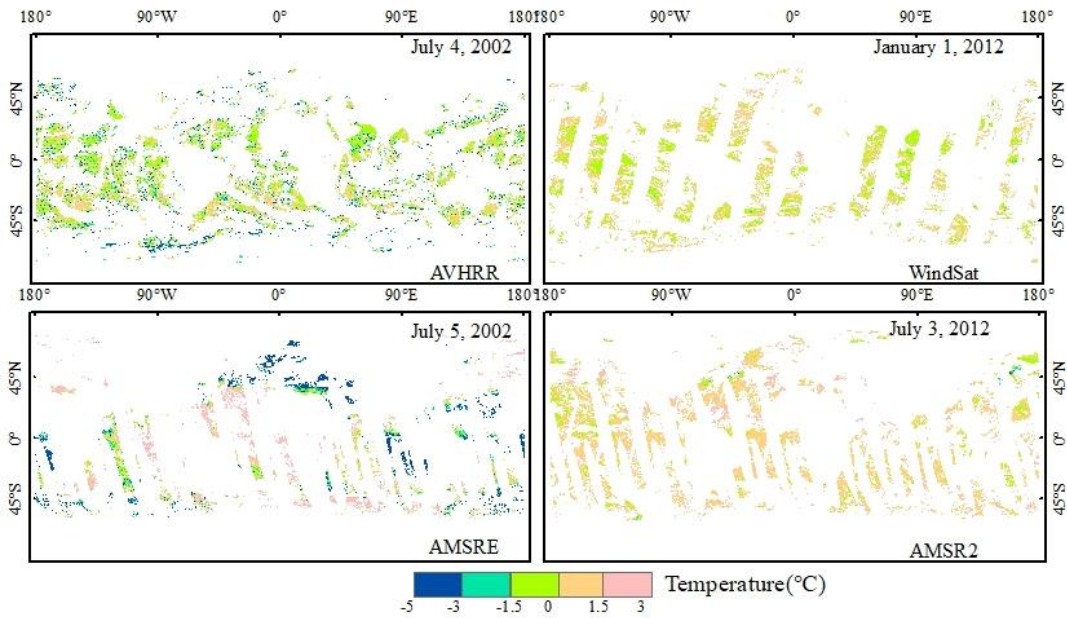



Figure 3. Difference maps of the original MODIS and multisource daily SST products. Areas of missing data are blank.

The main source of difference is mainly due to the inconsistent wavelengths or frequency ranges used by different sensors; these differences cause the sensors to obtain temperature information from different ocean depths. The sea temperature inversion algorithms of different sensors cause the measured temperatures to be higher/lower due to inconsistencies among key parameter settings, which cause the inversion results to be closer to the temperatures of the ocean surface or a given subsurface layer. Due to differences in the absorption of solar radiation, heat exchange with the atmosphere and levels of

subsurface turbulent mixing (Minnett et al., 2011), near-surface temperatures are highly variable vertically, horizontally and temporally (Minnett, 2003). Infrared remote sensors retrieve sea surface skin temperatures at depths of 10-20 µm. Microwave remote sensors retrieve sea surface subskin temperatures at 1-1.5 mm depths. Therefore, the SSTs retrieved from various microwave radiometers (AMSRE, WindSat, and AMSR2) are different from the SSTs measured by the MODIS radiometer. Although the AVHRR sensor is an infrared remote sensor and its brightness temperatures represent the sea

surface skin temperature, AVHRR SSTs correspond to subsurface SSTs because they are statistically regressed to coincident in situ buoy SSTs (Chao et al., 2009b; Kilpatrick et al., 2001; Pisano et al., 2016). Starting with the AVHRR Pathfinder Version 5.3, an average skin/subsurface temperature difference of 0.17 K, determined from Marine Atmospheric Emitted Radiance Interferometer (MAERI) matchups, was used to eliminate the subsurface bias so that the SSTs were more closely tuned to the sea surface skin temperatures (Sea Surface Temperature-Pathfinder C-ATBD). MODIS SSTs are skin SSTs.

MODIS retrievals are based on empirical coefficients derived by regressing MODIS brightness temperatures against in situ observations from drifting and moored buoys, but the regressed SSTs are converted to skin SSTs based on at-sea measurements. Thus, the SSTs retrieved from the AVHRR radiometer are different from the SSTs measured by the MODIS radiometer. In addition, MODIS and several other sensors used in this paper have different observation times and can obtain measurements at several different times throughout the diurnal cycle. The relationships among these observations are,

however, not constant because there are significant diurnal variations in sea surface temperature resulting from constant changes in the atmosphere, solar heating, wind speeds, etc. (Kilpatrick et al., 2015; Luo et al., 2019; Minnett et al., 2019; Wick et al., 2004). This also results in differences between MODIS observations and those of other sensors. Therefore, compensating for measurement depths and times is conducive to reducing the uncertainty present in the reconstruction results before the multisource remote sensing data are combined into the MODIS SST product.

1) Compensating to ensure uniform effective sampling depths

To solve the differences among MODIS and multisource daily SST products caused by the sampling depths, it is necessary to consider the differences as results of the cool skin effect and diurnal heating (Luo et al., 2020). Therefore, the model proposed by Fairall et al. (1996) was used to estimate the skin effect of infrared remote sensing products when integrating microwave remote sensing SSTs into infrared data, as shown in Eqs. 1 and 2:

$$\Delta T = Q\delta/K \qquad (1)$$

$$\delta = \frac{\lambda V}{\mu_{*w}} \qquad (2)$$



where $\Delta T$ is the temperature variation (positive, representing that the surface is cooler than the bulk), Q is the net heat flux, K is the thermal conductivity of water, $\delta$ is the thickness of the change in temperature, $\lambda$ is the empirical coefficient, V is the kinematic viscosity, and $\mu_{*w}$ is the friction velocity in the water. It is difficult to obtaining $\lambda$ in Eq. 2. Based on the

observed data of the Tropical Ocean-Global Atmosphere Coupled Ocean-Atmosphere Response Experiment (COARE) program, Fairall et al. (1996) determined $\lambda$ to be dependent on wind speed. General ocean models typically simulate the surface layer of 5-10 m as a uniform layer, and simulating such thin sea surface skin layers and subskin layers takes a long time. The General Ocean Turbulence Model (GOTM) can use a nonuniform grid and specifically encrypt the surface layer to quickly simulate the temperature of the sea surface skin layer and the subskin layer. The formula is as follows:


$$h_k = D \frac{\tanh\left((d_l+d_u)\frac{k}{M}-d_l\right)+\tanh(d_l)}{\tanh(d_l)+\tanh(d_u)} - 1 \tag{3}$$

where $h_k$ represents the thickness of layer K, D represents the depth, M is the number of layers, and $d_l$ and $d_u$ show the zooming factors of the surface and bottom, respectively.

From this formula, the following grids are constructed:

- dl = du = 0 results in equidistant discretization.
- dl > 0, du = 0 results in zooming near the bottom.
- dl = 0, du > 0 results in zooming near the surface.
- dl > 0, du > 0 results in double zooming near both the surface and the bottom.

In addition, the GOTM can be used to simulate the hydrodynamic and thermodynamic processes of vertical mixing in one-dimensional water columns in natural waters and can be used for depth corrections taking into account

atmosphere-ocean interactions and vertical turbulent mixing. Therefore, the Fairall model was integrated into the air-sea interaction module of the GOTM, and the heat and momentum flux changes of each layer in the water column were integrated to more accurately simulate the skin effects of the SSTs. In this section, the conversion of SSTs between different depths can be conducted using the model by entering the SST measurement depth and the corresponding meteorological parameter values present during the measurement, including the wind speed at a 10-m height, the air

temperature at a 2-m height above the sea surface, air humidity data, and cloud cover data from the ECMWF. Figure 4 (a) and (b) show the variations in ocean temperatures at different depths and the differences between the sea surface skin temperatures and sea surface subskin temperatures simulated by the GOTM every half hour for a pixel with a longitude of 32.65°N and a latitude of 43.25°E from July 1, 2002, to July 31, 2002. When the wind speed is low, the infrared-measured SST is 0.1~0.2℃ lower than that obtained by microwave remote sensing. When the wind speed is high, the SSTs

measured by the two sensor types are basically the same. By deducting this difference, the SSTs obtained by microwave remote sensing can be normalized to the SSTs obtained by infrared remote sensing.

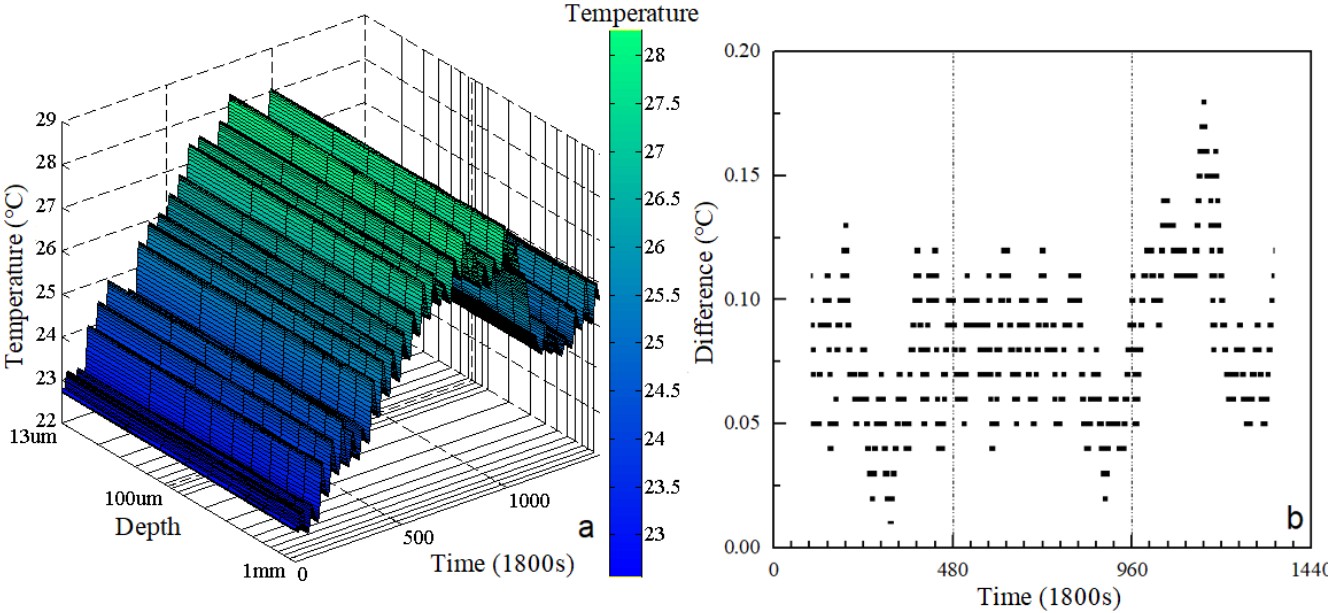

Figure 4. SST depth changes simulated by the GOTM every half hour for a pixel with a longitude of 32.65° and a latitude of
43.25° in July 2002 (a is the variation in ocean temperature at different depths; b is the difference between the sea surface
skin temperature and sea surface subskin temperature).

2)   Compensating to ensure uniform measurement times

To solve the differences among the MODIS and multisource daily SST products caused by the varying measurement times,
it is necessary to consider the diurnal variations in SST. The GOTM is based on the hydrodynamic and thermodynamic
processes of water and comprehensively considers the effects of solar shortwave radiation, longwave radiation, latent heat,
sensible heat and cloudiness on diurnal variations in SST. The diurnal variations caused by differences in the absorption and
attenuation of solar radiation of different water types are also considered. Therefore, the GOTM can accurately simulate
diurnal variations in SST. We use the GOTM to simulate diurnal variations in SST. The input data also come from the
ECMWF reanalysis product and include the wind speed at a 10-m height, the air temperature at a 2-m height above the sea
surface, air humidity data, and cloud cover data. Cloudiness is used to calculate oceanic radiant heating. Wind speed, air
temperature and relative humidity are used as inputs in the turbulence model to estimate sensible heat, latent heat and wind
stress. The exchange coefficient of the turbulence equation is obtained based on the Fairall parameter method. Figure 4 (a)
shows the variations in ocean temperature at different half-hour increments for a pixel with a longitude of 32.65° and a
latitude of 43.25° from July 1, 2002 to July 31, 2002. For the SSTs occurring at different times, after deducting the diurnal
variations in temperature simulated by the GOTM, the observations can be referenced to common time. The formula is as
follows:

$$SST_s = \frac{\sum_{i=1}^{N}(SST_s(i)+(SST_g(j)-SST_g(i)))}{N} \qquad (4)$$

where $SST_S$ is the SST observed by the satellite; j is the referenced common time; i is the effective observation of other moments by the sensor on the same day other than moment j, of which there are a total of N; and SSTg is the SST simulated by the GOTM, which also corresponds to moments i and j.

3)   Bias Adjustments of different sensor products

After completion of the above depth and diurnal change corrections, the different measurement times and effective sampling depths were compensated. However, the performances of different sensors are different, and there may be systematic and regional deviations, which need to be eliminated before fusion (Alerskans et al., 2020; Huang et al., 2015). Therefore, to correct the large-scale deviations among different sensors, we used the daily MODIS SST data to correct the

other remotely sensed data compensated for different measurement times and effective sample depths. Figure 5 shows that the correlation coefficient of the MODIS SSTs and the other remotely sensed data reaches above 0.97, indicating that these data have a strong correlation with the MODIS data. Therefore, we adopt linear regression to modify the other remotely sensed SST data. The correction method uses linear regression of two corresponding images, and the regression coefficient is determined by matching the data of the MODIS sensor and the other remotely sensed data. To avoid the influence of

individual outliers, points with standard deviations over 1°C or with a difference greater than 2°C from the corresponding MODIS datum in the matching window did not participate in the regression.

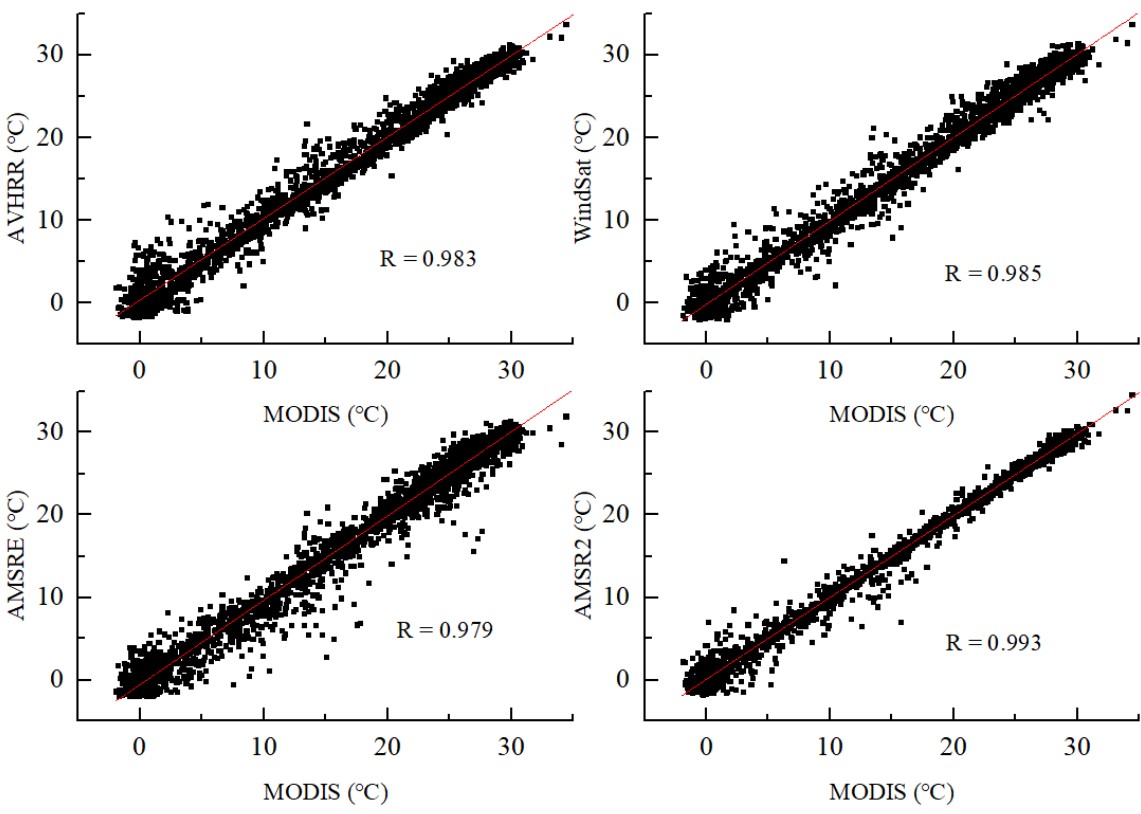

Figure 5. Scatter diagrams of the MODIS SST data and ocean multisource data compensated for different measurement times





and effective sampling depths.

### 2.4.1.2 Bias adjustment scheme for in situ observations

SSTs retrieved from MODIS sensors are skin SSTs. However, the in situ SSTs from Version 2.1 NOAA iQuam are subsurface SSTs. For Argo floats, only the shallowest high-quality measurement is extracted and saved from each profile into the iQuam dataset (the same algorithms are used for other in situ platforms, such as those on ships, drifters, and moorings), along with its measurement depth. The closest measurement to the surface of the Argo float is at a depth of 3-8 dbar (0.15-0.2 m for drifters and ~1 m for moorings). The differences between skin and subsurface SSTs, as described by Donlon et al. (2002), can be as large as 1.0°–2.0°C when the solar insolation is strong and the wind speed is weak. Figure 6 shows that the differences between the MODIS data and the eight types of in situ SSTs from iQuam can be significant under different weather conditions. When combining in situ SSTs into the MODIS SST product, such differences need to be accounted for. Therefore, in situ SSTs were first collocated and made coincident with MODIS data (within ±1 hr and ± 0.02° of latitude and longitude). Then, the coincident in situ SSTs were adjusted using the GOTM by entering the SST measurement depth and corresponding meteorological parameter values present during the measurement, including the wind speed at a 10-m height, the air temperature at a 2-m height above the sea surface, air humidity data, and cloud cover data from the ECMWF.

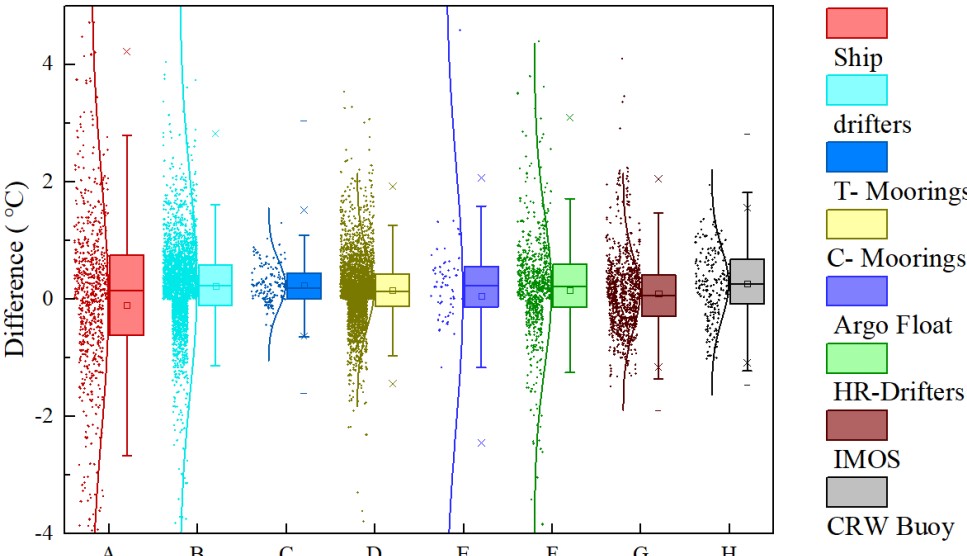

Figure 6. Box chart with scatters representing the differences between the original MODIS data and eight types of in situ SST observations.

### 2.4.2 Filtering of MODIS SST

The monthly MODIS SST data cover the whole sea area of the world, but they contain many missing and low-quality pixels caused by factors such as clouds and aerosols. Figure 7 shows the frequency of nonnull pixels, including valid pixels and

low-quality pixels, in the monthly MODIS SST data from July 2002 to December 2019. The missing pixels are mainly

distributed in high latitude sea areas beyond ±60 degrees of latitude. In the middle- and low-latitude sea areas within ±60 degrees of latitude, the coverage rate of pixels is more than 95%, among which missing pixels are mainly distributed in ocean edges near land. In most areas of low and middle latitudes, the nonnull pixel coverage is as high as 100%, but it is difficult to detect the cold top surface of thin clouds or subpixel clouds, and the SSTs retrieved under such conditions are usually underestimated because the temperatures of clouds are almost always colder than the temperature of the sea surface

(Reynolds et al., 2007). Moreover, other factors can also contaminate the observed signals and affect the data quality, such as factors related to the radiometer, including its viewing geometry, spectral response and noise level (Kilpatrick et al., 2015). Therefore, there are low-quality pixels present during the study period. In this study, the spatial process of the SST reconstruction includes the removal of low-quality pixels in low- and mid-latitude regions and the reconstruction of pixels in the marginal low- and mid-latitude regions and the high-latitude regions.

The quality control information stored in the qual_sst layer is provided along with the MODIS L3m SST data, with the quality level 0 being the best quality and the quality level 4 being the worst. These values can be found in the original MODIS SST Netcdf files (see section 2.1 for a detailed description). The missing pixels present in these data are represented by the filling value -32767. Therefore, the quality control labels and the filling value were used to identify low-quality and missing pixels in the MODIS SST product. For monthly and daily SST data, to ensure the data quality and the number of

effective pixels, pixels with a quality level ≤ 1 were considered to be high-quality data.

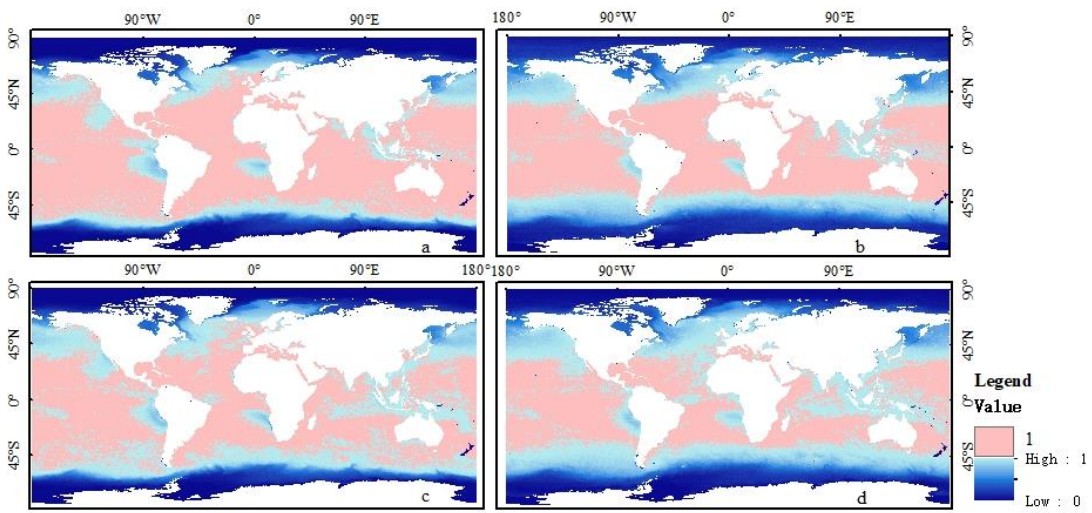

Figure 7. Frequency of nonnull pixels, including valid pixels and low-quality pixels, in the monthly MODIS SST data during the study period from (a) nighttime Aqua overpasses, (b) daytime Aqua overpasses, (c) nighttime Terra overpasses, and (d) daytime Terra overpasses.

**2.4.3 SST data reconstruction**

In the data processing, we first filtered all input monthly MODIS SST images and determined the locations of the low-quality and missing pixels. Then, for each invalid pixel (i.e., the low-quality and missing pixels) in the monthly images, we filtered the daily MODIS SST data of the respective month at the corresponding location. The high-quality pixels in the daily SST data were retained, and the invalid pixels in the daily data were reconstructed by combining multisource data.
Finally, the invalid pixels present in the monthly data were replaced by the mean SST values derived from the gap-filled daily SST time series of the corresponding month. Combining the characteristics of multisource data and the availability of the data, we adopted different methods to reconstruct the invalid pixels present in the daily MODIS SST data for different regions.

### 2.4.3.1 Reconstruction of invalid SST pixels in low- and mid-latitude marginal regions of the ocean

Due to the influence of the mixed sea and land pixels in adjacent coastal areas, microwave-based sea surface temperature products have very large uncertainties in adjacent coastal areas (Xie et al., 2008). Therefore, we first used daily SST data from MODIS and AVHRR and corresponding in situ observations to reconstruct the pixels in these regions. In cases where these observations were missing, we filled these invalid pixels based on the geographically weighted regression (GWR) and Kalman filtering (KF) methods, fitted the SSTs obtained by the two methods and finally reconstructed the invalid pixels. A
summary flowchart of the process is schematically illustrated in Figure 8.

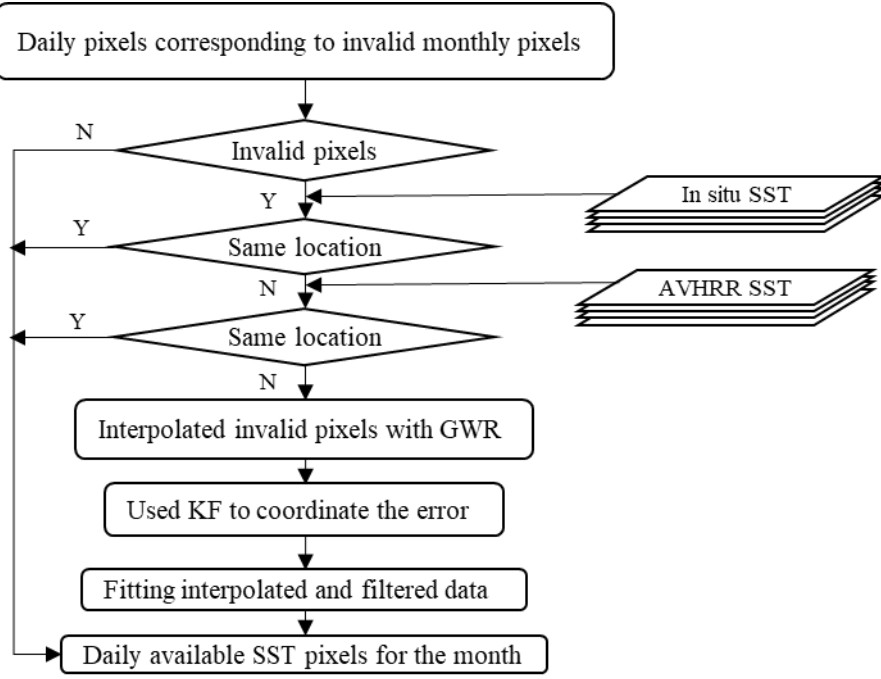

Figure 8. A summary flowchart for reconstructing invalid SST pixels in the marginal low- and mid-latitude regions
Invalid pixels were filled with in situ or AVHRR SST data at the same location and time (priority was given to the use of



in situ data), and these pixels filled with in situ observations were marked. For the invalid pixels without AVHRR and in situ
SSTs, we filled in the missing part with GWR method based on the slowly changing characteristics of SST, and then used
KF to coordinate the variation error and interpolation error of SST. Finally, the interpolation and filtered SST data were
fitted to realize the SST filling.

1)  Interpolating invalid pixels with GWR

GWR is an effective method for estimating missing pixels, and it can quantitatively determine the contribution of adjacent
pixels to contaminated pixels (Zhao et al., 2020). Therefore, the GWR method was used in this study to reconstruct invalid
pixels. To determine the sliding window with the minimum noise and the best complement value, we simulated the size of
the experimental pixel window several times and selected a sliding window of 5 by 5 pixels centered on the target pixel. This
window size also avoids the reduction in execution efficiency caused by the redundancy of pixel data involved in the
calculation and ensures the number of pixel values involved in the calculation. In theory, even in the case of thick cloud
coverage, in situ SSTs are the most reliable records. If there are in situ SST observations, the missing or low-quality pixels
are directly obtained from the in situ measurements, which are more representative of the real SSTs under cloud cover than
under clear sky conditions. During the reconstruction of invalid pixels, the regression weight coefficient of each adjacent
pixel was determined by the Euclidean distance between that pixel and the target pixel. Simultaneously, we assigned a
relative multiple weight to the marked in situ data according to GWR. By selecting some marked pixels as experimental
values, it was found that the target pixels can be estimated most accurately when $M_c$ ($M_c$ is the weighting coefficient of the in
situ assigned pixels) was set to 3 in this paper. The weighting coefficients of adjacent pixels can be determined by the
following formula. After the GWR model used the Euclidean distance to obtain the weights, a local linear regression
calculation was performed for each point in the window according to the sample weights. This regression calculation can be
expressed as Eq. 7:

$$D = \sqrt{(x - x_t)^2 + (y - y_t)^2} \tag{5}$$

$$W_i = \frac{\frac{M_c}{D_i}}{\sum_{i=1}^{m}\frac{M_c}{D_i} + \sum_{j=1}^{n}\frac{M_g}{D_j}}, W_j = \frac{\frac{M_g}{D_j}}{\sum_{i=1}^{m}\frac{M_c}{D_i} + \sum_{j=1}^{n}\frac{M_g}{D_j}} \tag{6}$$

$$T_t = \sum_{i=1}^{m} W_i \cdot T_i + \sum_{j=m+1}^{n} W_j \cdot T_j \tag{7}$$

where D is the distance from the adjacent pixel to the target pixel; (x, y),( $x_t$, $y_t$) are the locations of the adjacent pixel and
target pixel, respectively; i and j are the adjacent pixels used to estimate the SST of the invalid pixel; i is an adjacent pixel of
high quality; j is a pixel assigned by the in situ measurement; $W_i$ and $W_j$ are weight multipliers; m is the number of i; n is the
number of j; and $M_c$ and $M_g$ represent the weighting coefficients of the high-quality pixels and in situ assignment pixels,
respectively. In this paper, $M_c$ and $M_g$ are set at 1 and 3, respectively. $T_t$ is the filled SST value of the target pixel.

2)  Using KF to coordinate the error

For this region, on the basis of interpolation, KF can be used to coordinate the error characteristics of the SST variation
and the error characteristics of the interpolation. Since the SST variation is relatively flat, SST is treated as a stationary




random process. Due to the slowly changing characteristics of SST and the lack of in situ data representing these pixels, we took into account the observed data representing the three days before and after at the location of the invalid pixel. The MODIS products from the Terra and Aqua satellites produced 4 SST images per day, for a total of 28 images. Considering the operational requirements of SST real-time retrievals and the necessary computing speed and storage capacity of the

computer, the correlation of the error changes with each observation time was not considered in the actual operation process, and only simple random error was used to simulate the changes in the process error and observation error. By modeling the data, the equation of state of the system can be written as follows:

$$X_{(t)} = X_{(t-1)} + W_{(t-1)} \tag{8}$$

where X is the daily MODIS SST without interpolation; t and t-1 are time; and W represents the process noise, which is

considered to be Gaussian, and its covariance is represented by $Q_t$. Taking July 2002 as an example, there were 124 MODIS data points. All data were arranged in chronological order, and the change in each pixel relative to the previous time was counted. Based on the statistical results of these images, the mean square deviation of the change was 1.7648. Therefore, $Q_t$ is $1.7648^2 * I$ (I is the identity matrix).

Consider the following measurement equation:

$$Z_{(T)} = HX_{(t)} + V_{(t)} \tag{9}$$

where Z is the interpolated daily MODIS SST; H is an identity matrix; and V represents the measurement noise, which is also considered to be Gaussian, and its covariance is represented by $R_t$. $R_t$ is determined mainly by the accuracy of Z. In this paper, the covariance between the interpolated daily MODIS data and the in situ data collected during the study period was calculated to be 0.945; then, the following formula was used to combine the input data to achieve the optimal output of the

system.

The next state estimate was calculated using the state extrapolation equations.

$$X_{(t)}^{-} = X_{(t-1)} \tag{10}$$

The extrapolated estimate uncertainty (variance), $P_{(X(t)^-)}$, is the uncertainty of the extrapolated estimate.

$$P_{(X(t)^-)} = P_{(X(t-1))} + Q_{(t-1)} \tag{11}$$

The KF gain, $K_{(t)}$, was then calculated.

$$K_{(t)} = P_{(X(t)^-)} / (P_{(X(t)^-)} - R_{(t)}) \tag{12}$$

The current estimate was calculated using the state update equation.

$$X_{(t)} = X_{(t)}^{-} + K_{(t)}[Z_{(t)} - X_{(t)}^{-}] \tag{13}$$

The current estimate uncertainty was updated.

$$P_{X(t)} = [1 - K_{(t)}] P_{(X(t)^-)} \tag{14}$$

3) Fitting interpolated and filtered data

The least square method was used to fit the interpolated and filtered data, where in the interpolation based on the WGR method and the filtering based on KF were represented by $T_g$ and $T_k$ in the following equation, respectively.



$$T' = \alpha T_g + \beta T_k \tag{15}$$

To determine the coefficient of fitting, we selected some pixels from the images replaced by the in situ and AVHRR SSTs and marked them as invalid pixels and then interpolated and filtered these pixels. The least square method was used to fit the interpolated and filtered data, and the fitting coefficient was obtained using the following formula. Then, the reconstruction of invalid pixels without in situ or AVHRR SST filling could be realized by using Eq. 16:

$$\Delta T(\alpha, \beta) = \sum_{i=1}^{n} [T_i - T_i']^2 \tag{16}$$

where $T_i$ is the pixel value in the image replaced by the in situ and AVHRR SSTs and n is the number of these pixels. When $\Delta T$ reaches a minimum value, the fitting coefficient can be obtained by using Eqs. 17 and 18.

$$\frac{\partial \Delta T(\alpha, \beta)}{\partial \alpha} = -2 \sum_{i=1}^{n} (T_i - \alpha T_g - \beta T_k) T_g = 0 \tag{17}$$

$$\frac{\partial \Delta T(\alpha, \beta)}{\partial \beta} = -2 \sum_{i=1}^{n} (T_i - \alpha T_g - \beta T_k) T_k = 0 \tag{18}$$

**2.4.3.2 Reconstruction of invalid SST pixels in low- and mid-latitude inner ocean areas**

Similar to the method used to reconstruct invalid SST pixels in the marginal regions of the oceans at low and middle latitudes, pixels with invalid SST values were collocated with in situ and AVHRR data to reconstruct invalid pixels in inner ocean areas. The invalid pixels were filled using values from valid in situ SST or AVHRR data collected at the same location at the same time. The difference is that the accuracy of microwave-based data is lower in marginal sea areas but higher in the ocean interior. Based on the consistency of MODIS daily data and microwave daily data temperature variation trends at

corresponding dates, in cases of missing observations, we used microwave-based data to reconstruct the invalid SST data. A summary flowchart of the process is schematically illustrated in Figure 9.

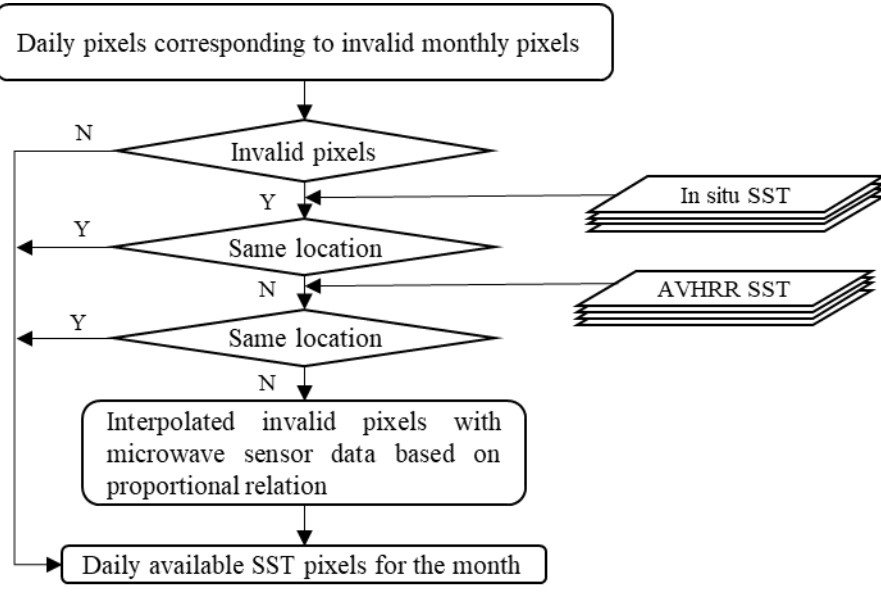

Figure 9.The summary flowchart for reconstructing invalid SST pixels in low- and mid-latitude inner ocean areas



The temperature variation trends present in the MODIS daily data and microwave daily data on corresponding date in the
same region are the same, so the two groups of data have the same proportional relation. Taking a grid of n pixels by n pixels as
an example, a and b are considered the same regions clipped from the MODIS and microwave-based data, respectively. The
gray and white rasters represent the effective and invalid pixels, respectively. $M_{kl}$ and $R_{kl}$ represent the pixel values of the
MODIS and microwave-based data, respectively. K and L represent the pixel positions. $M_{ij}$ represents the value after the
interpolation of invalid pixels.

Figure 10.    MODIS and microwave SST data corresponding to a n×n grid

$$\frac{M_{ij}}{\sum_{k=1,l=1}^{k=i-1,l=j-1} M_{k,l} + \sum_{k=i+1,l=j+1}^{k=n,j=n} M_{k,l}} = \frac{R_{ij}}{\sum_{k=1,l=1}^{k=i-1,l=j-1} R_{k,l} + \sum_{k=i+1,l=j+1}^{k=n,j=n} R_{k,l}} \tag{19}$$

The reconstruction of invalid pixels can be achieved by using the above formula. The reconstructed pixels meet the accuracy
of the interpolated images to a certain extent and do not damage the original SST variation trend of the interpolated image.
After several simulations of different experimental pixel window sizes, the noise was found to be minimized when a sliding
window of 6 by 6 pixels was used, and this window size was considered to have the best complement value.

**2.4.3.3 Reconstruction of invalid SST pixels in high-latitude regions of the ocean**

At high latitudes, sea ice covers a significant fraction of the global oceans (approximately 5-8%). The presence of large areas
of mixed sea ice and open water makes it difficult to retrieve SSTs (Høyer et al., 2012; Vincent et al., 2008). In addition,
there is persistent cloud cover in polar regions, with cloud cover occurring up to 90% of the time in summer and 50%-60%
of the time in winter in the Arctic (Høyer et al., 2012). The continuous cloud cover and extended twilight period complicate
the detection of cloud, which thus present problems for identifing clouds correctly of cloud detection algorithms. Therefore,
it is challenging to use satellite sensors to accurately retrieve SST at high latitudes, including the Arctic Ocean. Moreover,
because of the existence of sea ice and the difficulty of navigating in ice-filled water, the amount of field observations at the
area is generally scarce compared to other regions (Reynolds et al., 2002). The Microwave and AVHRR SST data used in
this study have limited available pixels in high latitude regions, so it is impossible to reconstruct MODIS SST data in high
latitude regions only by relying on these data and in situ data.





High-latitude SSTs can be estimated based on satellite sea ice concentrations (SICs). In areas with sea ice, the SST is the temperature of the open water or of the water under the ice (Banzon et al., 2020). Multiple analysis (L4) products from
GHRSST enable SST estimation near the polar region by converting SIC into SST. Due to differences in satellite source data, integration methods and methods for converting SIC to SST, the accuracy of levels 4 SST products of GHRSST-PP vary in many aspects. After understanding the differences among current GHRSST level 4 products and their qualities and availabilities in different areas, the OISST V2.1 product was selected to restore invalid pixels in the MODIS SST data in the high-latitude area with sea ice coverage. In the product, SICs were revised to SSTs to remove warm biases in the Arctic
region.

In areas of high latitudes, since the microwave-based SST data (used in this paper) exclude sea ice pixels, that is, SSTs are missing when the number of pixels with sea ice contamination exceeds a specified value, we used a combination of two strategies to reconstruct the missing SST data to improve the accuracy of the results. A summary flowchart of the process is schematically illustrated in Figure 11.

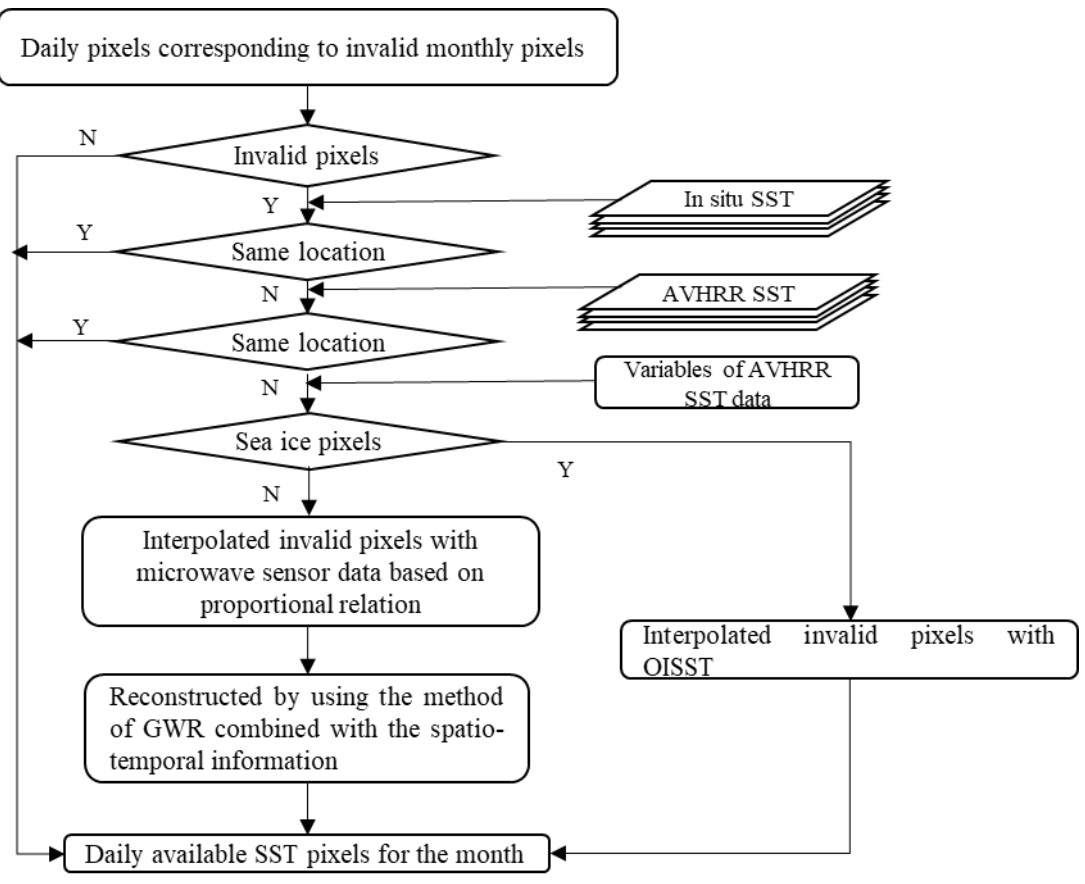


Figure 11. The summary flowchart for reconstructing invalid SST pixels in high latitude regions of the ocean

First, the variables l2p_flags and the sea ice fraction in the AVHRR SST data were used to identify the sea ice extent. The



sea ice fraction variable quantified the fraction of sea ice contamination in a given pixel (ranging from 0 to 1), and bit 2 of the l2p_flags variable was recorded if an input pixel recorded ice contamination. These variables can be used to identify sea
ice pixels. Then, we used the first strategy to reconstruct invalid pixels in high latitudes without sea ice coverage. Pixels with invalid SST values in the MODIS data were collocated with in situ and AVHRR observations. Invalid pixels were filled using the values from the valid in situ or AVHRR data at the same location and the same time (priority was given to the use of in situ data). Then, for the invalid pixels without available observations, we used the method described in section 2.4.3.2 above to fill the pixels using microwave data. Finally, considering the characteristics of the slow changes in SST and the fact
that SST changes in the same area are interannual and its changes in the short term are usually small, the invalid pixels without any filling data were reconstructed by using the GWR method combined with spatiotemporal information. That is, we obtained the effective pixels of the same date in the previous year and the following year at the same position and the effective pixel values of the adjacent dates (within two days) in the same year. Then, the invalid pixels were replaced with the composite average pixel value. If the number of effective pixels was too small, then the GWR method was used to
reconstruct the invalid pixel.

In another strategy, grid cells with invalid SST values due to sea ice-covered areas were collocated with in situ and AVHRR SSTs. Invalid pixels were filled using values from valid in situ SST or AVHRR data collected at the same location and at the same time. Then, the adjusted OISST V2.1 products were used to reconstruct the invalid pixels in sea ice-covered areas that did not have sufficient replacement pixels. The adjustment algorithm is a linear regression algorithm that relies on
coefficients derived from collocated, cotemporal OISST and MODIS SST observations. These data were then used to fill the missing value pixels by linear interpolation:

$$T_M = \alpha \times T_O + \beta \tag{20}$$

where $T_M$ is the SST after interpolation (units: °C); $T_O$ is the pixel value of the OISST product; $\alpha$ is the correction factor of the OISST on the SST value from MODIS, which is the regression coefficient based on the clear sky pixels of daily MODIS
SSTs in the corresponding two images; and $\beta$ is the estimated intercept.

**3 Result**

MODIS has superior coverage and performance in sampling global SST and has been verified by various studies (Barton and Pearce, 2006). Moreover, to better assess the accuracy of the new SST product, we performed verification of the original MODIS data, oceanic multisource data compensated for different measurement times and effective sampling depths, and the
new SST data in different regions. The accuracy of the data was assessed using five statistical indexes: the correlation coefficient ($R^2$), root mean squared error (RMSE), bias, absolute bias (Abs_Bias), and scatter index (SI). The bias was calculated as the MODIS SST product minus the in situ SST. The scatter index, usually denoted as SI, was used to measure the magnitude of the bias between the SST product and the in situ observations versus the in situ observations. A smaller SI means a more accurate measurement. In addition, to convey information more easily and concisely, Taylor diagrams (Taylor,



2001) were also used to compare the accuracies of different SST products, as they provide a way to graphically summarize the relative accuracies of several products. Taylor diagrams are two-dimensional scatter plots in which discrete points give an indication of how well patterns match each other in terms of their correlation coefficient (R), centered RMSE (E), and normalized standard deviation (SDV), all at once (Castro et al., 2016). These statistics are defined as follows, where M and O are the simulated and observed patterns, respectively.

$$R = \frac{1}{N-1}\sum_{i=1}^{N}\left(\frac{m_i - m}{\sigma_m}\right)\left(\frac{o_i - o}{\sigma_o}\right) \tag{20}$$

$$SDV = \frac{\sigma_m}{\sigma_o} \tag{21}$$

$$E^2 = \frac{(RMSE^2 - bias^2)}{\sigma_o} \tag{22}$$

$$E^2 = SDV^2 + 1 - 2SDV \times R \tag{22}$$

In the Taylor diagram, SDV is shown as the radial distance, and R is shown as the cosine of an azimuthal angle in the
polar plot. The observed patterns are represented by points on the X-axis at R = 1 and SDV = 1. E is the distance from the simulated patterns to the observed patterns, and this distance can quantify how closely the simulated patterns resemble the observed patterns.

### 3.1 Evaluation of the original product

We conducted a comparative analysis based on the distribution of invalid pixels in different regions, as shown in Tables 1
and 2. The Arctic Ocean was not verified because the original data had many missing pixels at high latitudes. Tables 1 and 2 show the validation results of the original monthly MODIS SST values against the in situ SST measurements and the in situ SST measurements compensated for the effective sampling depths, respectively. Without correction of the sampling depths of the in situ SST measurements, the MODIS-based daytime SST measurements showed positive biases, while the MODIS-based nighttime SST measurements showed negative biases except over the Atlantic Ocean. With corrected
sampling depths of the in situ SST measurements, MODIS-based nighttime SST measurements of the Atlantic Ocean also showed negative biases. In addition, MODIS-based daytime SST products are better than nighttime products, and the measurements over the Atlantic Ocean have the lowest accuracies.

Table 1. Analysis of SST matching points between original monthly MODIS-TERRA/AQUA and in situ SST measurements from 2002 to 2019

| | Day/Night | $R^2$ | Abs_bias | bias | RMSE | SI |
|---|---|---|---|---|---|---|
| Pacific Ocean | d | 0.9773 | 0.6693 | 0.2240 | 1.1513 | 0.0548 |
| | n | 0.9786 | 0.6807 | -0.1821 | 1.1665 | 0.0532 |
| Atlantic Ocean | d | 0.9561 | 0.8279 | 0.2543 | 1.4527 | 0.0709 |
| | n | 0.9584 | 0.9634 | 0.0921 | 1.6225 | 0.0970 |





|  | Day/Night | $R^2$ | Abs_bias | bias | RMSE | SI |
|---|---|---|---|---|---|---|
| Indian Ocean | d | 0.9786 | 0.6929 | 0.0271 | 1.1606 | 0.0498 |
|  | n | 0.9836 | 0.6082 | -0.2009 | 1.2412 | 0.0613 |
| Global Ocean | d | 0.9721 | 0.7262 | 0.1900 | 1.2593 | 0.0591 |
|  | n | 0.978 | 0.7079 | -0.1434 | 1.2666 | 0.0615 |

Table 2. Analysis of SST matching points between original monthly MODIS-TERRA/AQUA and in situ SST measurements adjusted by GOTM from 2002 to 2019

|  | Day/Night | $R^2$ | Abs_bias | bias | RMSE | SI |
|---|---|---|---|---|---|---|
| Pacific Ocean | d | 0.9850 | 0.5794 | 0.1528 | 0.9134 | 0.0437 |
|  | n | 0.9880 | 0.5995 | -0.2360 | 0.8925 | 0.0397 |
| Atlantic Ocean | d | 0.9776 | 0.6822 | 0.1015 | 0.8254 | 0.0503 |
|  | n | 0.9821 | 0.7624 | -0.1709 | 1.0686 | 0.0622 |
| Indian Ocean | d | 0.9892 | 0.5654 | 0.0150 | 0.8254 | 0.0406 |
|  | n | 0.9954 | 0.5072 | -0.2924 | 0.7162 | 0.0326 |
| Global Ocean | d | 0.9843 | 0.6097 | 0.1054 | 0.9314 | 0.0439 |
|  | n | 0.9898 | 0.6031 | -0.2393 | 0.8849 | 0.0414 |

## 3.2 Evaluation of the bias adjustment

### 3.2.1 Evaluation of satellite data bias adjustment

Different sensors and satellites can obtain measurements at several different times throughout the diurnal cycle. In addition, microwave and infrared sensors have different effective measurement depths. Since both the AMSRE and MODIS instruments are aboard the AQUA satellite, they both pass through the equator at approximately 01:30 and 13:30. Therefore, to verify the depth compensation conducted by the GOTM, we used the GOTM to perform depth correction on the daily AMSRE data and then compared the corrected values with the corresponding MODIS daily data collected at the same time. Figure 12 (a) shows the validation results of the AMSRE data sampling depth compensated by the GOTM. It can be seen that the corrected data have better consistency, with the RMSE value being reduced from 1.137 to 0.508 and the absolute bias being reduced from 0.718 to 0.302, indicating that the GOTM can simulate the SST at different depths well and can be used for SST conversions between different depths. Furthermore, we compared and analyzed the nighttime products of the same sensor with the corresponding daytime products after a time correction to verify the time correction performed by the GOTM. Taking AMSRE daytime products that pass through the equator at approximately 13:30 as an example, we corrected the values to AMSRE nighttime products that pass through the equator at 01:30 (Figure 12 (b)). With this method, the region that experiences a temperature increase from 01:30 to 13:30 is well-corrected. Comparing the SST values before and after the correction with the actual SST at 01:30, there was an obvious daily temperature increase before the correction, and the



data after the correction had lower absolute bias and RMSE values, indicating that the GOTM can simulate the diurnal variation in the SST well and can be used to normalize the SSTs observed at different times.

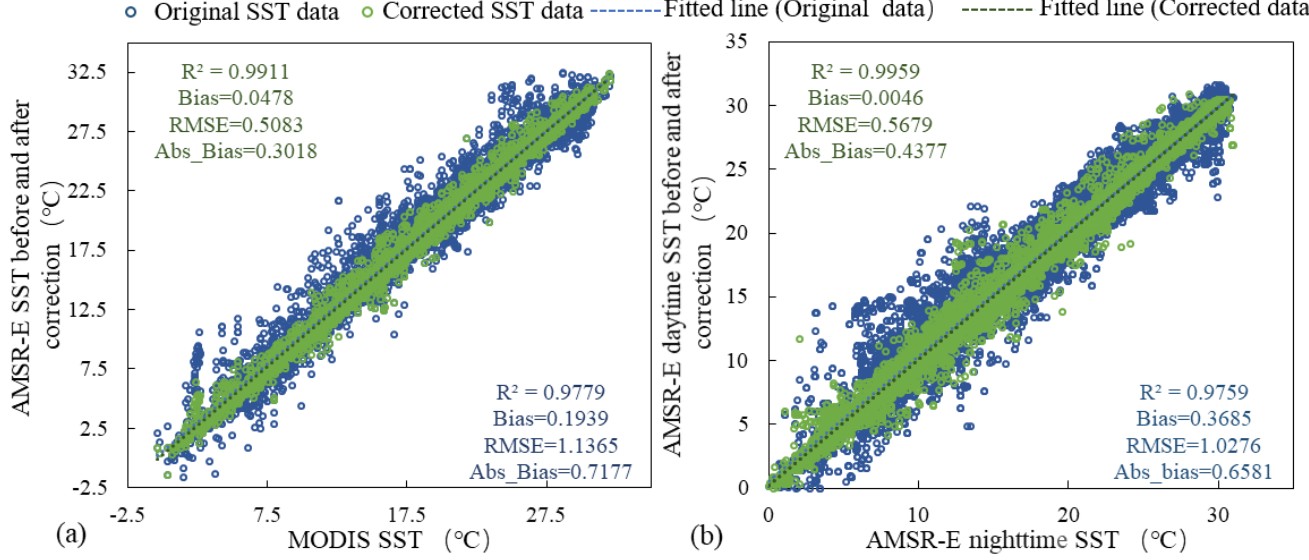

585

Figure 12. The scatter diagrams of the daily original SST data and corrected results versus their corresponding actual SST data from 2002 to 2019. The blue points indicate original SST pixel values. The green points represent the values in corrected SST data, and the statistical accuracy measures ($R^2$, Bias, Abs_Bias, and RMSE) are also indicated.

**3.2.2 Evaluation of in situ data bias adjustment**

590 To validate the results of using the GOTM for the depth compensation of in situ SSTs, we selected the matchups corresponding to the effective pixels of daily MODIS SSTs from the in situ SSTs and compared and analyzed the daily MODIS SSTs with these matchups corrected by the GOTM. Figures 13 and 14 show the verification results of the MODIS SSTs against the in situ data before and after the calibration, respectively. Figure 13 reflects the change in the difference between all types of in situ data before and after the correction and the corresponding MODIS SST data. The SSTs from the 595 MODIS sensor and in situ observations showed a large deviation without the depth compensation, and the deviation was significantly reduced after the correction. Figure 14 is based on the MODIS SST as a reference and shows the distribution of SSTs before and after the correction from the 8 platforms described in the normalized Taylor diagram. In Figure 14, the degrees of agreement are compared among the in situ data from different platforms before and after the correction with the MODIS data. The points representing the in situ SSTs lying near the MODIS observations (the MODIS observations are 600 represented by points on the X-axis at R = 1 and SDV = 1) have relatively high R and low E values. After the depth correction, the points representing the in situ SSTs are closer to the MODIS observations, which means that compared with the in situ data before correction, the agreement between the two is better. Therefore, the corrected result of the GOTM is stable and reliable and can be used for the conversion of SSTs from in situ observations taken at different depths.



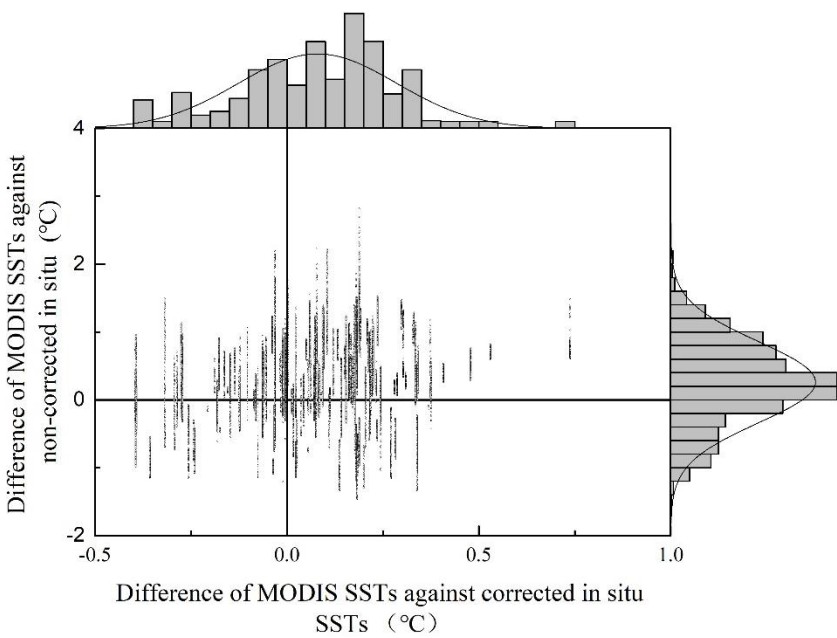

605 Figure 13. Marginal Histogram of the difference between in situ data before and after correction and the corresponding MODIS SST data. (The margins of the scatterplot is a histogram of the variables, indicating the distribution of data in either direction)

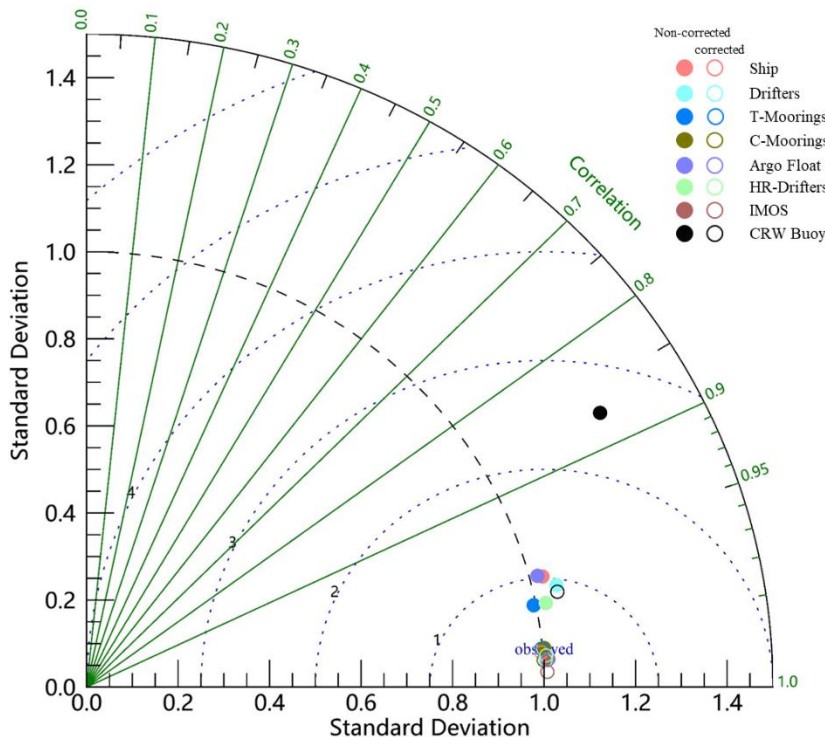



Figure 14. Normalized Taylor diagrams showing differences between matched SST from in situ data before and after
correction and the corresponding MODIS SST data.

## 3.3 Evaluation of the new product

### 3.3.1 Accuracy verification of low-quality pixels

In this study, we only restored invalid pixels, including low-quality pixels and missing pixels, in the MODIS data and first
evaluated the improvement effect of these pixels. Figure 15 shows the validation results of the low-quality MODIS SST data
and the reconstruction results versus the corresponding in situ observations, including the corrected and uncorrected data,
showing the comparison of the accuracies of the low-quality pixels before and after the reconstruction. The validation results
show that the reconstructed MODIS SST data are always more consistent with the in situ data, including the corrected data
and uncorrected data, than the values before reconstruction, with RMSE values lower than 0.675 and R values higher than
0.991. Compared with the original values, the accuracies of the corrected values are improved by more than 0.65°C.

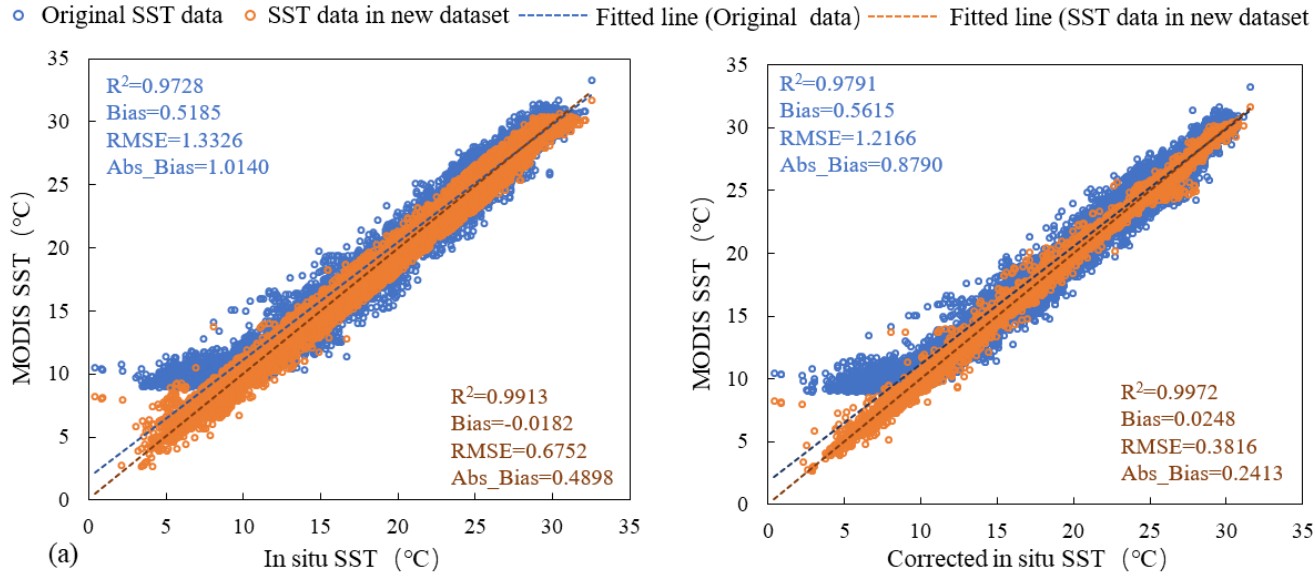

Figure 15. The scatter diagrams of the low-quality MODIS SST data and the reconstruction results versus their
corresponding in situ SST data from 2002 to 2019. The blue points indicate low-quality MODIS SST pixel values. The
orange points represent the values in reconstructed SST data, and the statistical accuracy measures ($R^2$, Bias, Abs_Bias, and
RMSE) are also indicated.

### 3.3.2 Overall accuracy verification

To fully verify the overall accuracy of the reconstructed SST products, we compared the performances of the original
MODIS SST and the reconstructed SST products relative to the in situ dataset via Taylor diagrams. The normalized Taylor

diagrams showing the performances of the two products relative to the in situ data before and after the correction are presented in Figure 16. Compared with the original MODIS product, the reconstructed product can better represent the in situ observations, with the highest R value, lowest E value and SDV closest to one. Among them, the original MODIS product with the lowest consistency with both the uncorrected observations and the corrected observations by far consists of the Atlantic SSTs, with E=0.090 and 0.057, SDV=0.967 and 0.982, and R=0.954 and 0.9711, respectively. After reconstruction, the Atlantic SSTs show very good correlation, with a lower E value and SDV closer to 1 with both the corrected and uncorrected observations, and its accuracy is significantly improved.

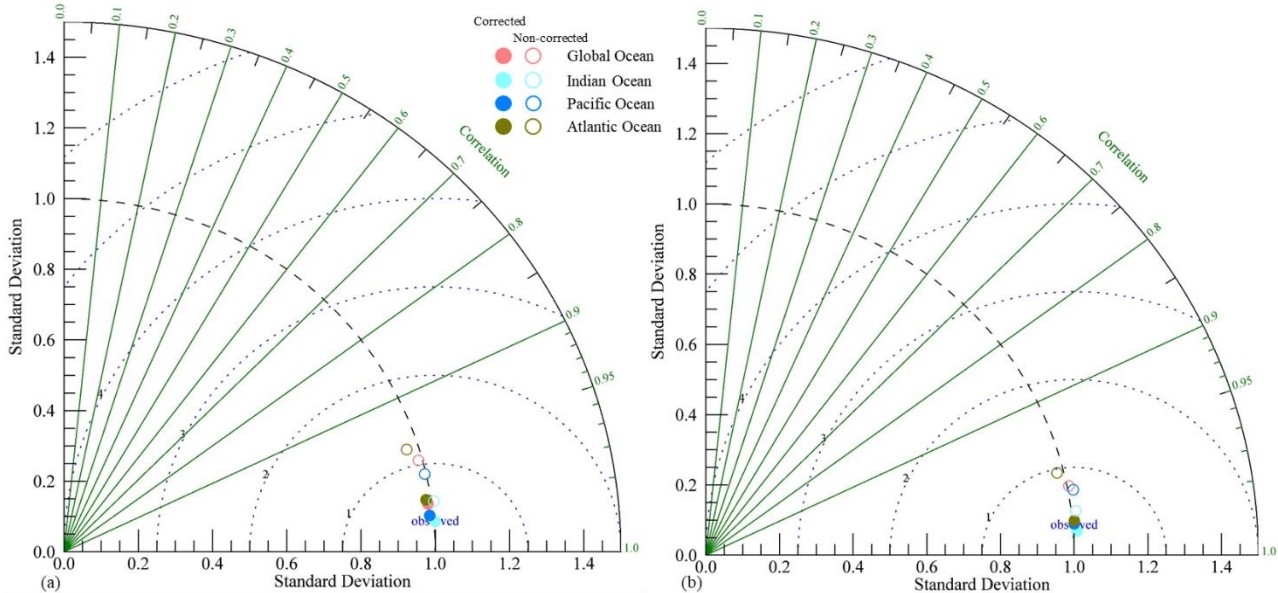

Figure 16. Normalized Taylor diagrams showing differences between matched SST from in situ data before (a) and after (b) correction and the corresponding SST products.

To further understand the credibility of the reconstructed product and clarify the limitations of this method, we further assessed the performance in terms of the output biases in different regions. The associated validation statistics of the new SST dataset against the corrected in situ observations and uncorrected in situ observations are summarized in Table 3. The new dataset is in agreement with the uncorrected in situ observations with abs_bias=0.3358, RMSE=0.5767, and SI=0.0352 on the global ocean. Among these statistics, the RMSE, SI and abs_bias of the Atlantic region are slightly larger than the values in the global ocean, but they are all better than those of the original MODIS SST data (see Table 1 for details), the correlation coefficients of this product in different regions are all greater than 0.984, and the SI is less than 0.004. The abs_bias of the new SST product relative to the corrected in situ observations is 0.3349°C, and the RMSE and SI are 0.4742 and 0.0242, respectively. The RMSE, SI and abs_bias of the values of the Atlantic Ocean region are also slightly larger than those of the global values. However, they are still better than those of the original MODIS SST data (see Table 2 for details), the correlation coefficients of the product in the different areas are greater than 0.995, and the SI is less than 0.0032. In





addition, the RMSE and SI values of the edge areas and high latitude areas are slightly lower than the global values, which
indicates that the accuracy of the data in these areas is higher. These results indicate that the reconstructed MODIS SST
dataset is robust and accurate due to its high consistency with in situ observations, including corrected and uncorrected
observations. Therefore, we believe that the accuracy of SST data can be improved by the method adopted in this paper.

Table 3 Statistics of the validation results of new SSTs against in situ SST measurements (non- corrected/corrected)

|  | In situ data | $R^2$ | Abs_bias | RMSE | SI |
|---|---|---|---|---|---|
| Pacific Ocean | Non-c | 0.9888 | 0.2977 | 0.5219 | 0.0306 |
|  | corrected | 0.9960 | 0.3226 | 0.4618 | 0.0219 |
| Atlantic Ocean | Non-c | 0.9846 | 0.4343 | 0.7657 | 0.0391 |
|  | corrected | 0.9952 | 0.3666 | 0.4864 | 0.0320 |
| Indian Ocean | Non-c | 0.9963 | 0.3095 | 0.5010 | 0.0238 |
|  | corrected | 0.9977 | 0.2529 | 0.4080 | 0.0186 |
| Global Ocean | Non-c | 0.9906 | 0.3358 | 0.5767 | 0.0352 |
|  | corrected | 0.9961 | 0.3349 | 0.4742 | 0.0242 |
| Arctic Ocean | Non-c | 0.9933 | 0.3660 | 0.5161 | 0.0298 |
|  | corrected | 0.9971 | 0.3122 | 0.4738 | 0.0243 |
| marginal regions | Non-c | 0.9941 | 0.3360 | 0.5049 | 0.0269 |
|  | corrected | 0.9983 | 0.3342 | 0.467 | 0.0219 |

To investigate the performance of the reconstructed product relative to the other products, a comparison between the
OISST product and the reconstructed data in this study was conducted during 2002-2019. OISST Version 2.1 is an analysis
product constructed by combining observations from different platforms on a regular global grid, such as AVHRR data from
NOAA satellites, ships, Argo float and drift floats, with a spatial grid size of 0.25°. For the OISST images, we averaged the
daily SST data corresponding to each month and obtained monthly SST images. Then, the dataset was validated against the
corresponding in situ observations, including the uncorrected and corrected in situ SSTs, as shown in Figure 17 (a). The
RMSE values of OISST against the uncorrected and corrected in situ observations in the global ocean were 0.602°C and
0.495°C, respectively. Those of the reconstructed SSTs against the uncorrected and corrected in situ observations in the
global ocean were 0.577°C and 0.474°C, respectively. Compared to these, the overall accuracy of the reconstructed data is
better. In addition, we also performed an intercomparison with the 2° Extended Reconstructed Sea Surface Temperature
(ERSST) product, which is a global monthly SST dataset derived from the International Comprehensive Ocean–Atmosphere
Dataset (ICOADS) that uses statistical methods to enhance spatial completeness. Figure 17 (b) reflects the monthly average
SST changes in different oceans from the ERSST product and the reconstructed products over the 2002-2019 period,
indicating a reasonable consistency between the two. Based on the accuracy assessment and data intercomparison results, it
can be seen that the reconstructed MODIS products of 2002-2019 are reliable with high accuracies and that the reconstructed



models we designed are effective.

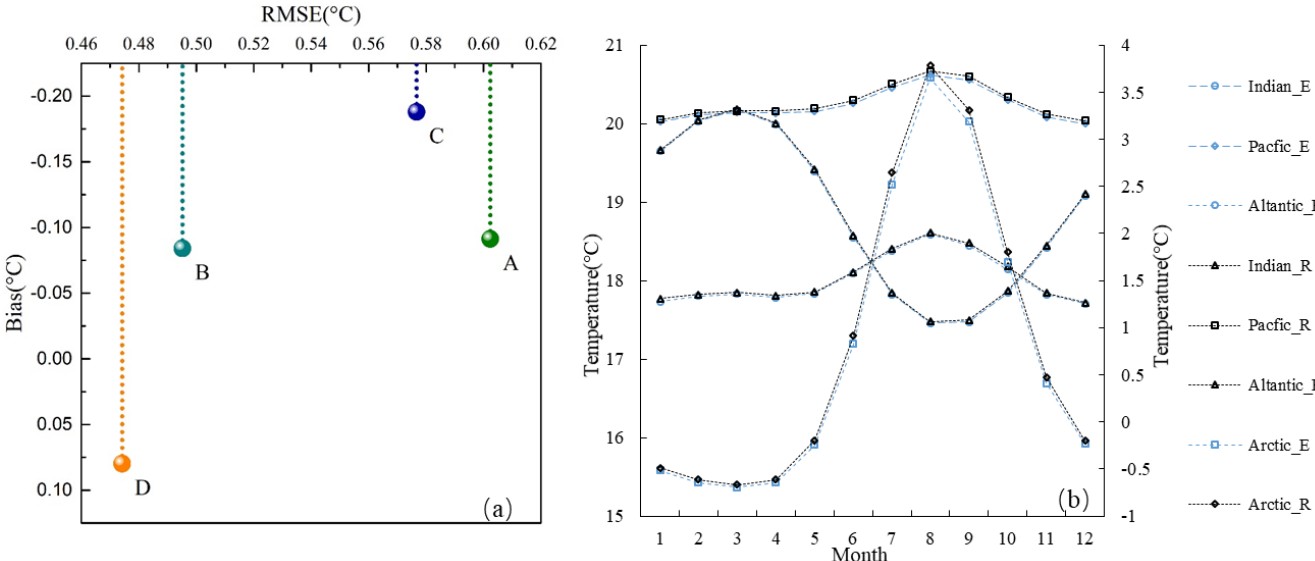


Figure 17. Validation statistics of the reconstructed product and other SST products during 2002–2019. (a): intercomparison with OISST, A and C represent the results of OISST and new SSTs against uncorrected in situ SST measurements, respectively. B and D represent the results of OISST and new SSTs against corrected in situ SST measurements, respectively. (b) intercomparison with ERSST, black and blue are monthly mean SST changes of ERSST and new SSTs in different ocean

from 2002 to 2019.

## 4 Discussion

SST dataset with high accuracy, spatial completeness and fine resolution has important research and application value in the research of global change, disaster prevention and mitigation, and economics. The complementary information of the SST data derived from multiple satellite sensors in spatial completeness and accuracy makes it possible to generate the improved

global coverage, high- quality unified SST data set by integrating the multiple SST products.

There are many differences in terms of effective sampling depths and measurement times of SST products derived from various instruments, which will lead to complicated temporal and spatial differences in SST of different products (Castro et al., 2004; Minnett et al., 2011; Wick et al., 2004). Therefore, the diurnal variations of SST at different surface depths must also be considered in the merging of multi-source data. The GOTM model simulates the hydrodynamic and thermodynamic

processes of vertical mixing of one-dimensional water columns in natural waters, and comprehensively considers the effects of solar short-wave radiation, long-wave radiation, latent heat, sensible heat and cloudiness on the diurnal variations of SST, which can more accurately simulate the diurnal variations of SST than traditional empirical regression models that only consider the main factors of diurnal variations (such as wind speed, solar radiation, etc.). In addition, it has a high vertical



resolution and can be encrypted on the surface layer to simulate the difference between the skin layer and the sub-skin layer,
so as to achieve the uniformity of temperature at different observation depths. Therefore, the GOTM simulation method was
used to unify the temporal and spatial reference of SST at different depths and different times for each pixel of the image,
and the accuracies of each sensor and in situ observations are improved by 0.3-0.8℃. However, there are still certain errors,
which are not only related to the characteristics of different sensors, retrieval algorithms, etc., but also to the accuracy of the
GOTM model simulation. The simulation accuracy of GOTM largely depends on the input meteorological parameters. The
wind speed, sea temperature, relative humidity, cloud cover and other data used in this paper come from ECMWF reanalysis
and forecast data. The spatial resolutions of these data are relatively low, and the temporal resolutions are 3-6 hours, which
are obviously insufficient for the rapidly changing volume such as wind speed and cloud cover. If the meteorological
parameters with higher accuracy and resolution are available, the simulation accuracy is expected to be improved. In addition,
when correcting the in situ observations from the iQuam, not every in situ observations from iQuam record the depth at the
time of measurement. For example, the actual depth measured by the drift buoy is not fixed at 0.2m, which will fluctuate due
to the action of waves and so on. Therefore, there will be a certain deviation in the correction to the skin layer, and these
factors will ultimately affect the accuracy of the reconstructed product.

In addition, the SST data in the grid form represent the average temperature in the grid area, while the in situ observations
represent just the temperature near the locations of the station. Although this study uses the average value of the high-quality
observations that fall in the grid area with temporal sampling less than or equal to 1 h as the matched data of the grid, it is
still limited by the number of measured data within the grid. Especially in the high latitude areas where the measured points
are sparse, the uncertainties associated with such matches could potentially bias the reconstruction and validation results.
Therefore, more meteorological observation stations will be need to help improve the accuracy of the product. The
acquisition and integration of rasterized SST is a complex problem, and the reconstruction models proposed in this research
is just the beginning, which needs to be improved and developed continuously. How to better solve the time phase and
sampling depth problems of satellite remote sensing data, and to introduce multiple types of data sources into the model is a
way to improve the product accuracy, which needs further in-depth research in the future.

## 5 Data availability

The Reconstructed MODIS SST products at 0.041° resolution from 2002 to 2019 are freely available to the public in the
img format at http://doi.org/10.5281/zenodo.4419804 (Cao et al., 2021), which are distributed under a Creative Commons
Attribution 4.0 License.

## 6 Conclusions



This study presents a new SST product with full spatiotemporal coverage based on multisource data after calibration by using a temperature depth and observation time correction model. The product, generated by inputting infrared-based, microwave-based and in situ SST data into the reconstruction spatial model, has a monthly temporal interval and a 0.041° spatial interval. This dataset effectively removed approximately 25% of the missing pixels or low-quality SST pixels from original MODIS monthly images. Detailed comparisons and analyses with the in situ observations (including uncorrected in situ data and corrected in situ data) and OISST and ERSST products illustrate the reliability and accuracy of the reconstructed product. This dataset effectively addresses the issues of inconsistent observation times and sampling depths of multisource data and compensates for the insufficiency of reconstructing actual SST pixels under clear-sky conditions rather than under clouds in some studies with very limited information, achieving good temporal and spatial coverages; thus, this product can be used for mesoscale ocean phenomenon analyses. It will be of great use in research related to global change, disaster prevention and mitigation, and economic research. Moreover, the reconstruction strategy used in this study can be extended to the reconstruction of temporal and spatial gap-free fields of other multisource and multitemporal satellite data, providing technical support for the generation of satellite reconstruction SST series products with a unified spatiotemporal reference for any temporal and spatial intervals.

**Author contributions.** MC and KM designed the research and developed the methodology; MC wrote the manuscript; and KM, YY and all other authors revised the manuscript.

**Competing interests.** The authors declare no conflicts of interest.

**Acknowledgments.** The authors would also like to thank the National Aeronautics and Space Administration (NASA), the NOAA National Centers for Environmental Information, the Naval Research Laboratory (NRL) Remote Sensing Division and the Naval Center for Space Technology, and other agencies for their support by providing the SST product. We also thank the ECMWF for providing the climate reanalysis data. This work was supported by the National Key Project of China (Nos. 2018YFC1506602, 2018YFC1506502), Fundamental Research Funds for Central Non-profit Scientific Institution (Grant No. 1610132020014), and Open Fund of State Key Laboratory of Remote Sensing Science (Grant No. OFSLRSS201910).

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
