# Peer review of "A New Global Gridded Sea Surface Temperature Data Product Based on Multisource Data"

_Earth System Science Data, 2021_

## Author Comment (AC1)

**Dear Reviewers and Editor,**

This research was done with the support of China's National Key R&D Program. The purpose of the project is to build a long-term series of global major meteorological disaster remote sensing data sets with high spatio-temporal and consistency based on the current global multi-source remote sensing data and ground observation site data, and to provide key ocean temperature parameters (such as sea surface temperature) for marine meteorological disaster forecasting models, especially rapid forecasts of marine disasters such as typhoons, and provide early warning services for global fishing vessels and merchant ships.

We would like to express our sincere appreciation to Anonymous Referee #1 for his/her comprehensive comments and valuable suggestions on our manuscript which are very important to improve the quality of our manuscript. All the comments made by Anonymous Referee #1 have been carefully and individually addressed. The reply is in supplement.

**# Summary:**

This is a review of "A New Global Gridded Sea Surface Temperature Data Product Based on Multisource Data" by Mengmeng Cao, Kebiao Mao, Yibo Yan, Jiancheng Shi, Han Wang, Tongren Xu, Shu Fang, and Zijin Yuan. The authors have merged the sea surface temperature (SST) data from multiple sources to create a new highresolution global dataset of monthly SST. The new data product does not contain any missing values and has been shown to be more accurate than the unmerged datasets. However, I have some concerns about the usefulness of the new dataset and the assessment of accuracy (see general comments). Moreover, the authors may need to emphasize the novelty and the uniqueness of the methods used to create the new data set.

**Response:** We would like to thank you for reviewing our manuscript. Your comments and good suggestions are very important for us to improve the quality of manuscript and dataset. We have carefully addressed all the issues raised by you and the response is presented below.

At present, there are three main methods for obtaining ocean temperature: The first is the traditional method, which obtains sea surface temperature through sea observation sites. The main advantage of this method is that it has continuity in time and is hardly affected by weather. The disadvantage is that the number of observation sites is limited and the space lacks continuity, especially in remote sea areas. The second method is to obtain sea surface temperature through remote sensing retrieval. Remote sensing has advantages in space, but lacks continuity in time. Remote sensing is divided into two inversion methods: thermal infrared remote sensing and passive microwave remote sensing. Thermal infrared inversion of sea surface temperature has a high accuracy and resolution, but it has a great influence on clouds. There are 60% of the area covered by clouds every day, so there are 60% of the area missing values. Although passive microwave is less affected by clouds, the resolution is relatively low. Passive microwave remote sensing is affected by the land near the coast, and the accuracy of sea surface temperature inversion is not high. The third is to output sea temperature products through the assimilation model. This method relies on the accuracy of the input parameters of the assimilation model.

Although different methods are used to obtain ocean surface temperature, they actually represent temperature information at different ocean depths, and the observation time is also inconsistent. The sea temperature observed by traditional sites is deeper than the temperature observed by remote sensing. Even if they are all the temperatures retrieved from remote sensing, the temperatures retrieved from thermal infrared and microwave are from different ocean depths. The sea temperature observed by thermal infrared is the skin temperature, and the sea temperature observed by microwave is a bit deeper than the depth observed by thermal infrared. The sea surface temperature obtained by the assimilation model should also be different.

Thermal infrared remote sensing is currently recognized as the most accurate method for obtaining sea surface temperature in a large area. Therefore, thermal infrared remote sensing is usually used to obtain sea surface temperature. For the ocean, the effective sea surface temperature value obtained through thermal infrared remote sensing every day is less than 40% of the total area (as shown in Figure 1), which means that 60% of the daily data have no value every day.

When calculating the monthly average data, some data sets use the average temperature value obtained by dividing the effective days of a month by the effective days. For example, if a certain pixel has only 25 effective temperature values in a certain month, then the average temperature of this month is calculated by using the average value of these 25 valid days. Although the monthly average temperature map calculated in this way has few missing values, the average temperature error of some pixels is relatively large (as shown in Figure 2).

The highlight of our work is to make full use of the research foundation of the predecessors to traverse the MODIS ocean temperature data set to find the pixels with low quality data, and then use high-quality daily data and other multi-source data (including sea surface temperature retrieved from passive microwave and observation site data, etc.) to improve the accuracy of the data, including pixels with low data quality and missing pixel data.

---

## Author Comment (AC2)

Dear Reviewers and Editor,

The purpose of this work is to build a long-term series of global SST dataset with high spatio-temporal and consistency based on the current global multi-source remote sensing data and ground observation site data with the support of China's National Key R&D Program ,which can provide key parameter for marine meteorological disaster forecasting models, especially rapid forecasts of marine disasters such as typhoons, and provide early warning services for global fishing vessels and merchant ships.

We thank you for your comprehensive comments and guidance and good suggestion. All the comments have been carefully and individually addressed. Enclosed below are our point to point responses to these comments.

The manuscript describes the methodology and procedure adapted to produce a new global dataset with 0.041° spatial resolution of monthly SST fields. The authors use MODIS SST data as benchmark and many other supplementary and complementary data sets including in situ observations and those retrieved from AVHRR infrared sensors, and AMSR and Windsat microwave sensors are utilized for obtaining a fusion. Essentially, the values in the blank or missing and low quality pixels are replaced by values of in situ observations and those derived or interpolated through the processes of Optimal Interpolation and Kalman Filter. The missing and low quality pixel problems arise due essentially to three reasons: (1) Cloudiness, fog, sea ice and proximity to shore influence the SST measurement. (2) Different sensors have different responses, that is, different sensors observe a pixel at different hours of the day and sense the temperature at different depths. (3) Latitudinal position of the pixel and the angle of sight from the sensor. The improvements in the new dataset, in comparison with earlier dataset, are statistically quantified.

**Response:** We would like to thank the referee for reviewing our manuscript. These comments are very important for us to improve the present manuscript. We have carefully addressed all the issues raised by the referee. Please find our detailed reply below.

A reader who is not highly specialized in the fields (of remote sensing and statistical manipulation of geophysical data) finds the manuscript difficult to read and assimilate. There is a certain amount of repetition in the description which did not contribute to clarity.

**Response:** Thank you for the valuable guidance. We have deleted these repetitive statements and revised some sentences that are difficult to understand.

The methodology section should be improved to offer more clarity. In many places, a lot of empiricism is found about parameters that cannot be measured directly or easily or with sufficient accuracy. To make the understanding easier, they should provide units for the variables in the equations, tables and figures. Also the figure legends need to be more complete.

**Response:** Thank you for your guidance, and we have tried to make revisions.

For Climatologists and Oceanographers who wish to use the SST, without bothering to go into miniscule details of the elaborate processing procedure, the present product provides a more accurate dataset. The quality control statistics presented shows substantial improvements in the new SST product.

One fundamental question over monthly time scales: Do we require a spatial resolution of 4.1 km, especially in the open oceans? This high resolution SST perhaps helps coastal studies like upwelling and estuary biology.

**Response:** Thank you for your positive evaluation and guidance. Your evaluation is very correct. The high-resolution ocean surface temperature data set is far more important for coastal studies than for open oceans studies. But the high-resolution ocean temperature data set is also helpful for us to capture temperature anomalies in open oceans, helping us to understand the ocean more accurately, such as the migration of central location of El Niño or La Niña.

On the whole the authors did a good and useful job. Some specific points to be considered are:

1. How can GOTM produce accurate values while utilizing several variables, such as 2 m temperature, 10 m wind, sensible heat flux, latent heat flux, which have poor accuracy? Does the ECMWF reanalysis of these variables present the accuracy needed?

**Response:** Thank you for your comment and guidance. You are right. When the accuracy of the input variables cannot be guaranteed, the GOTM simulation value will have deviations, such as 2 m temperature, 10 m wind, sensible heat flux, latent heat. In general, the use of ECMWF reanalysis of these variables can meet the requirements. But in special circumstances, when the input parameter deviation is too large, it will also cause a relatively large error. When there is a large deviation, we will make adjustments. For example, the difference between the sea temperature depth obtained by microwave inversion and the temperature depth obtained by thermal infrared inversion is less than 1mm, and the temperature difference is within 0.6 K, and we have tried to control this error through statistical methods.

2. Fig. 7: Why the non-null pixel frequency is low off the Peru coast, exactly in the ENSO signal region? Because, the region is covered by low cloud almost all the time. You can see that the ITCZ region and other tropical oceanic regions west of the continents also present low non-null frequency due to clouds, high as well as low clouds. Inclusion of these comments may enrich your manuscript.

**Response:** Thank you very much for your guidance and good suggestion. We have tried to make revisions.

3. Make the difference between skin temperature and surface temperature clear.

**Response:** Thank you very much for your careful review. We have modified it in the manuscript.

4. I agree that the differences in time and depth of observation have to be compensated. What guarantees that Eqs. 1 and 2 can fix these problems? l is empirical, m is the frictional velocity in the water. These parameters may introduce uncertainties. What is the sanctity of the formula in Eq. 3? As you said in the Discussion section, the procedure relies on the performance of GOTM.

**Response:** Thank you for the valuable comment. You are very right. Some parameters cannot be obtained in real time, especially in some remote areas. Some parameters in the formula are empirical in Eqs. 1 and 2, which will cause certain uncertainty. The reanalysis data such as 2 m temperature, 10 m wind, sensible heat flux, latent heat flux, etc. are used as input parameters for control, so the uncertainty of calculating the sea surface temperature will be controlled within a certain range, which can basically meet the current requirements. In addition, there is also microwave inversion temperature as a control condition. We try to improve the accuracy of sea surface temperature products as much as possible.

In our research, Eq. 3 is mainly used to determine the representative depth of the temperature observed by different observing instruments for stratification. Some parameters, such as reanalyzed data as input parameters, can still bring certain errors when the accuracy deviation is relatively large. Therefore, we need more and more high-precision input parameters, and this is also one of the main reasons why many countries have been continuously increasing the number of ocean ground observation sites and improving the spatial and temporal resolution of satellite observations.

5. Why can't you use the diurnal variability from in situ observations, at different places and in different months, instead of relying on a model?

**Response:** Thank you for the valuable comment. You are right. We originally planned to do this as you have said at the beginning, but we found that the number of data from the observed observation sites was very limited. Another important reason is that the data depth information of the observation sites is also inconsistent with the MODIS thermal infrared observation depth. We can still need to use the model for calibration. Therefore, we directly chose the model for calibration.

6. Lines 428-429: You say "I" is identity matrix and immediately after you say "H" is an identity matrix.

**Response:** Thanks a lot for pointing these out. We have modified it in the manuscript.

7. The procedure described in lines 430 through 458 needs some clearer explanation.

**Response:** Thank you very much for your careful review. We have modified it in the manuscript.

8. Tables: You better provide in the text expressions for the statistical metrics shown the tables.

**Response:** Thank you for your guidance. We have made revisions.

9. Figs. 12 and 13 call for authors' comments. Fig. 16: what do blank circles and filled circles represent? Fig. 17: Indian_R and Indian_E. Tell what they represent in the legend.

**Response:** Thank you for your guidance. We have made revisions.

10. L 683: By the expression "different surface depths" you mean "different depths in the surface layer"?

**Respons**e: Thank you for your guidance. We are sorry for our unclear expression. It refers to "different depths in the surface layer". We have made revisions.

11. Many uncertainties you mentioned in your discussion will remain uncertain for a long time to come. Rewrite the last sentence, L 700. At many other places too the write-up needs improvement. Some repetitions can be suppressed while more explanation and comments are needed in some places. The conclusion section can be merged with discussion section and some repetitions can be avoided.

**Response:** Thank you for your guidance. We have tried our best to improve the quality based on your opinion.

Please pay attention to comment 7 above.

**Response:** Thank you for the valuable comment and guidance. We have made revisions.

---

## Author Response (AR1)

Dear editor and referees,

Thank you for your valuable comments on our manuscript. First, we would like to express our sincere appreciation for your professional and insightful remarks on our paper. These comments are all valuable and have helped us to improve the quality of our paper. We have studied each comment and have made revisions that we hope will meet with approval. Please find our detailed responses below. For convenience, we also attach a version of the manuscript with changes incorporated. Thanks again.

With our best regards, Mengmeng Cao and co-authors

**Response to referees**

**Response to referee #1**

**# Summary:**

This is a review of "A New Global Gridded Sea Surface Temperature Data Product Based on Multisource Data" by Mengmeng Cao, Kebiao Mao, Yibo Yan, Jiancheng Shi, Han Wang, Tongren Xu, Shu Fang, and Zijin Yuan. The authors have merged the sea surface temperature (SST) data from multiple sources to create a new high-resolution global dataset of monthly SST. The new data product does not contain any missing values and has been shown to be more accurate than the unmerged datasets. However, I have some concerns about the usefulness of the new dataset and the assessment of accuracy (see general comments). Moreover, the authors may need to emphasize the novelty and the uniqueness of the methods used to create the new data set.

**Response:** We would like to thank you for reviewing our manuscript. Your comments and good suggestions are very important for us to improve the quality of manuscript and dataset. We have carefully addressed all the issues raised by you and the response is presented below.

At present, there are three main methods for obtaining ocean temperature: The first is the traditional method, which obtains sea surface temperature through sea observation sites. The main advantage of this method is that it has continuity in time and is hardly affected by weather. The disadvantage is that the number of observation sites is limited and the space lacks continuity, especially in remote sea areas. The second method is to obtain sea surface temperature through remote sensing retrieval. Remote sensing has advantages in space, but lacks continuity in time. Remote sensing is divided into two inversion methods: thermal infrared remote sensing and passive microwave remote sensing. Thermal infrared inversion of sea surface temperature has a high accuracy and resolution, but it has a great influence on clouds. There are more than 60% of the area covered by clouds every day, so there are more 60% of the area missing values. Although passive microwave is less affected by clouds, the resolution is relatively low. Passive

microwave remote sensing is affected by the land near the coast, and the accuracy of sea surface temperature inversion is not high. The third is to output sea temperature products through the assimilation model. This method relies on the accuracy of the input parameters of the assimilation model.

Although different methods are used to obtain ocean surface temperature, they actually represent temperature information at different ocean depths, and the observation time is also inconsistent. The sea temperature observed by traditional sites is deeper than the temperature observed by remote sensing. Even if they are all the temperatures retrieved from remote sensing, the temperatures retrieved from thermal infrared and microwave are from different ocean depths. The sea temperature observed by thermal infrared is the skin temperature, and the sea temperature observed by microwave is a bit deeper than the depth observed by thermal infrared. The sea surface temperature obtained by the assimilation model should also be different.

Thermal infrared remote sensing is currently recognized as the most accurate method for obtaining sea surface temperature in a large area. Therefore, thermal infrared remote sensing is usually used to obtain sea surface temperature. For the ocean, the effective sea surface temperature value obtained through thermal infrared remote sensing every day is less than 40% of the total area (as shown in Figure 1), which means that more than 60% of the daily data have no value every day.

When calculating the monthly average data, some data sets use the average temperature value obtained by dividing the effective days of a month by the effective days. For example, if a certain pixel has only 25 effective temperature values in a certain month, then the average temperature of this month is calculated by using the average value of these 25 valid days. Although the monthly average temperature map calculated in this way has few missing values, the average temperature error of some pixels is relatively large (as shown in Figure 2).

The highlight of our work is to make full use of the research foundation of the predecessors to traverse the MODIS ocean temperature data set to find the pixels with low quality data, and then use high-quality daily data and other multi-source data (including sea surface temperature retrieved from passive microwave and observation site data, etc.) to improve the accuracy of the data, including pixels with low data quality and missing pixel data.